# Heterogeneous decision making towards mixed autonomy: When uncertainty-aware planning meets bounded rationality

## Abstract

The past few years have witnessed a rapid growth of the deployment of automated vehicles (AVs). Clearly, AVs and human-driven vehicles (HVs) will co-exist for many years, and AVs will have to operate around HVs, pedestrians, cyclists, and more, calling for fundamental breakthroughs in AI designed for mixed traffic to achieve mixed autonomy. Thus motivated, we study heterogeneous decision making by AVs and HVs in a mixed traffic environment, aiming to capture the interactions between human and machine decision-making and develop an AI foundation that enables vehicles to operate safely and efficiently. There are a number of challenges to achieve mixed autonomy, including 1) humans drivers make driving decisions with bounded rationality, and it remains open to develop accurate models for HVs' decision making; and 2) uncertainty-aware planning plays a critical role for AVs to take safety maneuvers in response to the human behavior. In this paper, we introduce a formulation of AV-HV interaction, where the HV makes decisions with bounded rationality and the AV employs uncertainty-aware planning based on the prediction on HV's future actions. We conduct a comprehensive analysis on AV and HV's learning regret to answer the questions: 1) *How does the learning performance depend on HV's bounded rationality and AV's planning*; 2) *How do different decision making strategies impact the overall learning performance*? Our findings reveal some intriguing phenomena, such as Goodhart's Law in AV's learning performance and compounding effects in HV's decision making process. By examining the dynamics of the regrets, we gain insights into the interplay between human and machine decision making in mixed autonomy.

## 1 Introduction

Automated vehicle (AV) is emerging as the fifth screen in our everyday life, after movies, televisions, personal computers, and mobile phones Yurtsever et al. (2020); Parekh et al. (2022). The anticipated benefits from AV technology are immense, especially in terms of safety and economic impact Talebpour & Mahmassani (2016); Wu et al. (2017); Hoogendoorn et al. (2014); Ye & Yamamoto (2018). For example, in 2014, the National Highway Traffic Safety Administration (NHTSA) estimated the annual economic loss and society harm of crashes in the United States alone at $871 billion in 2010, or 1.9% of the GDP. Incredibly, the overwhelming majority of the crashes are preventable. As more technologies continue to deliver new safety and efficiency features to modern vehicles, the advent of AVs equipped with a myriad of sensors and AI technology has ushered in a new era of smart mobility. While 30+ states in the US have already enacted AV legislation, experts agree that the new phase of rapid global development of AVs must overcome a wide range of technical challenges Yuen et al. (2021); Jing et al. (2020); Litman (2020). In particular, maintaining safety while still being sufficiently efficient in a mixed-traffic environment is probably the most fundamental challenge for automated mobility. Indeed, AVs will have to operate around human-driven vehicles (HVs), pedestrians, cyclists, motorcyclists, and more, for many years to come. The complicated interactions between HVs and AVs could have significant implications on the traffic efficiency given their different decision making characters. As such, a fundamental understanding on the heterogeneous decision making in the interplay, especially the impact of HVs' decision making with bounded rationality on AVs' performance, is crucial for achieving efficient mixed autonomy.

Existing works on modeling the interaction between AV and HV largely fall within the realm of conventional game formulation, in which both agents try to solve the dynamic game and adopt Nash equilibrium strategies Tian et al. (2022); Hang et al. (2020); Fisac et al. (2019); Sadigh et al. (2016). This line of formulation faces the challenge of prohibitive computational complexity Daskalakis et al. (2009). Needless to say, the decision making of HV and AC are different by nature. As supported by evidence from psychology laboratory experiments Simon (1979); Kahneman (2003); Kahneman et al. (1982), human decision-making is often *short-sighted* and deviates from Nash equilibrium due to their *bounded rationality* in the daily life Selten (1990); Kalantari et al. (2023); Wright & Leyton-Brown (2010). In particular, HV's bounded rationality is unknown a prior and it remains challenging to develop an accurate model for HV's decision making. As a result, it is sensible for AVs' decision making to leverage *uncertainty-aware planning* for safety maneuvers in response to human behavior Liu et al. (2017); Schwarting et al. (2019). Clearly, the heterogeneous decision making by HVs and AVs exposes intrinsic complexities in the mixed autonomy.

Along the line of Sadigh et al. (2016; 2018), we consider a two-agent system with one AV and one HV, where the HV takes the action by planning for a short time horizon, and the decision-making is sub-optimal and noisy due to bounded rationality. The AV utilizes uncertainty-aware lookahead planning based on predictions of the HV's future actions. The primary objective of this study is to understand the performance of heterogeneous decision making in the mixed autonomy by answering the following questions: *1) How does the learning performance depend on HV's bounded rationality and AV's planning? 2) How do different decision making strategies between AV and HV impact the overall learning performance?*

The main contributions of this paper can be summarized as follows:

**(1) We first focus on the characterization of the regrets for both the HV and the AV, based on which we identify the impact of *bounded rationality* and *planning horizon* on the learning performance.** In particular, we present the upper bound on the regret, first for the linear system dynamics model case and then for the non-linear case. We start with the linear case, and show the accumulation effect due to the AV' prediction error and its impact on AV's learning performance. Building on the insight from the linear case, we model the prediction error as a diffusion process in the non-linear case to capture the accumulation effect. By studying the upper bound, we identify the compounding effects in HV's decision making due to bounded rationality and the Goodhart's law in AV's decision making associated with the planning horizon.

**(2) We study the impact of HV's bounded rationality on the overall learning performance and the regret dynamics of AV and HV.** We first establish the upper bound on the regret of the overall system due to HV's bounded rationality and AV's uncertainty-aware planning. Our regret bound naturally decompose into two parts, corresponding to the decision making of AV and HV, respectively. We examine the regret dynamics of the overall system theoretically and show how do different learning strategies between AV and HV affect the learning performance during each individual interaction through empirical study. The experiments details are available in Appendix H.

### 1.1 RELATED WORK

**Mixed Autonomy.** Previous studies on mixed autonomy generally consider specific dynamics models for AV and HV. For instance, Zhu & Zhang (2018) uses Bando's model to describe the AV and HV's behavior and demonstrates the traffic flow empirically on car-following model. Mahdinia et al. (2021) conducts a empirical study on the impact of AV on HV's performance in terms of the driving volatility measures while assuming a specific AV's acceleration model. Meanwhile, the impact of humans in the mixed traffic is empirically examined through high-fidelity driving simulator Sharma et al. (2018). Zheng et al. (2020) proposes a stochastic model for mixed traffic flow to investigate the interaction between HV and AV while taking into account the uncertainty of human driving behavior. It is shown that AV has huge impact on the overall traffic stability and HV's behavior through numerical study. Wu et al. (2017) proposes a modular learning framework for mixed traffic by leveraging deep RL. Experiments show that AV is able to reduce congestion under the intelligent driver model (IDM). Without imposing specific models on HV and AV's decision making dynamics, our work focuses on the performance of different learning strategies in the mixed autonomy.

**HV-AV Interaction Model.** In related work on modeling the HV-AV interaction, Tian et al. (2022) uses the general-sum Stackelberg game to account for the human's influence on AV and computes the backward reachability tube for safety assurances in the interaction. Similarly, Sadigh et al.

(2016) formulates the interaction as a two-player game, where the influence between AV and HV are captured in the predefined reward. Considering the vehicles' dynamic driving actions, Fisac et al. (2019) develops a hierarchical game-theoretic planning scheme and shows the effectiveness of the proposed planning method in the simulation. We note that even though Sadigh et al. (2018) proposes to use underactuated dynamical system to overcome the limitations of the game formulation, it assumes that both AV and HV making decision in the same strategy, i.e., planning for the same horizon. The related ad-hoc team problem Mirsky et al. (2022) mainly focused on the cooperative case, whereas our setting does not impose assumptions on the cooperation of agents. Meanwhile, Zero-shot coordination Hu et al. (2020) mainly focused on the robustness of the self-play, whereas our work aims to understand the interaction between two agents with different decision makings strategies. Moreover, our work focuses on characterizing the impact of the opponent modeling errors on the learning performance which is also related to opponent modeling Albrecht & Stone (2018). Despite the rich empirical results in the related filed, e.g., Ad-hoc team problem and zero-shot coordination, we remark that the theoretical analysis on the interaction between AV and HV is still lacking, especially considering their different decision making. Moreover, our work deviates from the conventional game setting and aims to takes steps to quantify the impact of AV and HV's different decision making on the traffic system.

**Model-based RL.** Model-based RL (MBRL), which leverages a model of the environment, is promising for real-world applications thanks to its data efficiency Moerland et al. (2023). In particular, our work is relevant to MBRL with lookahead planning. For instance, Sikchi et al. (2022); Xiao et al. (2019) use lookahead policy to rollout the dynamics model into the future $H$ steps in order to find the action sequence with highest return. A value function is also attached at the end of the rollout to estimate the terminal cost. Moreover, Sikchi et al. (2022) provides the sub-optimality gap of the learned policy under an approximate model and approximate value function. Our work is different from previous work on MBRL since in our case, AV has access to the environment dynamics while the modeling error exists due to the unknown bounded rationality of HV. Meanwhile, our theoretical analysis uses regret to evaluate the performance of the decision making, in which the value function is updated during the learning process, resulting in the changing function approximation error. As a result, the technique used in our proof is significant different than previous work Xiao et al. (2019); Sikchi et al. (2022); Luo et al. (2022).

## 2 PRELIMINARY

**Stochastic Game.** We consider the Stochastic Game (SG) defined by the tuple $\mathcal{M} := (\mathcal{X}, \mathcal{U}_A, \mathcal{U}_H, P, r_A, r_H, \gamma)$ Shoham & Leyton-Brown (2008), where $\mathcal{U}_A$ and $\mathcal{U}_H$ are the action space for AV and HV, respectively. In this work, we assume the action space for HV and AV are with the same cardinality $M$. For simplicity, we use $\mathcal{U} = \mathcal{U}_A \times \mathcal{U}_H$. We denote $\mathcal{X}$ as the state space that contains both AV and HV's states. $P(x'|x, u_A, u_H) : \mathcal{X} \times \mathcal{U} \times \mathcal{X} \to [0, 1]$ is the probability of the transition from state $x$ to state $x'$ when AV applies action $u_A$ and HV applies action $u_H$ and $r_H(x, u_A, u_H) : \mathcal{X} \times \mathcal{U} \to [0, R_{\max}], r_A(x, u_A, u_H) : \mathcal{X} \times \mathcal{U} \to [0, R_{\max}]$ is the corresponding reward for HV and AV. $\gamma \in (0, 1)$ is the discount factor. We denote the AV's policy by $\pi : \mathcal{X} \times \mathcal{U}$. Furthermore, we use $\hat{u}_H(t)$ to represent AV's prediction on HV's real action $u_H(t)$ at time step $t$. We use $\rho_0$ to represent the initial state distribution. Furthermore, we compare our problem formulation and Dec-POMDP in detail in Appendix B.

**Value Function.** Given AV's policy $\pi$, we denote the value function $V^\pi(x) : \mathcal{X} \to \mathbb{R}$ as

$$\mathbf{E}_{u_A(t)\sim\pi, x(t+1)\sim P(\cdot|x(t), u_H(t), u_A(t))} \left[ \sum_{t=0}^\infty \gamma^t r_A(x(t), u_A(t), u_H(t))|x(0) = x, u_H(t) \right],$$

to measure the average accumulative reward staring from state $x$ by following policy $\pi$. We assume the maximum value of the value function to be $V_{\max}$. We define $Q$-function $Q^\pi(x, u_A, u_H) : \mathcal{X} \times \mathcal{U} \to \mathbb{R}$ as $Q^\pi(x, u_A, u_H) = \mathbf{E}_\pi[\sum_{t=0}^\infty \gamma^t r_A(t)|x(0) = x, u_A(0) = u_A, u_H(0) = u_H]$ to represent the expected return when the action $u_A, u_H$ are chosen at the state $x$. The objective of AV is to find an optimal policy $\pi^*$ given HV's action $u_H$ such that the value function is maximized, i.e.,

$$\max_\pi \mathbf{E}_{x\sim\rho_0}[V^\pi(x)] = \max_\pi \mathbf{E}_{x\sim\rho_0, u_A\sim\pi(\cdot|x, u_H)}[Q^\pi(x, u_A, u_H)]. \tag{1}$$

**Notations.** We use $\|\cdot\|$ or $\|\cdot\|_2$ to represent the Euclidean norm. $\|\cdot\|_F$ is used to denote Frobenius norm. $\mathcal{N}(\mu, \sigma^2)$ is the normal distribution with mean $\mu$ and variance $\sigma^2$. $I$ is an identity matrix.

## 2.1 Modeling AV-HV Interaction: Heterogeneous Decision Making

In this section, we examine in detail the interaction between one AV and one HV in a mixed traffic environment. More specifically, we have the following models to capture the interplay between human and machine decision making in the mixed autonomy.

**AV's Decision Making via $L$-step lookahead planning.** At time step $t$, after observing the current state $x(t)$, AV will first need to predict HV's future action $\hat{u}_H(t+i), i = 0, 1, 2, \cdots, L-1$ due to the unknown bounded rationality of HV. Based on this prediction, AV strives to find an action sequence that maximizes the cumulative reward with the predicted HV actions using trajectory optimization. In order to facilitate effective long-horizon reasoning, we augment the planning trajectory with a terminal value function approximation $\hat{Q}_{t-1}$, which is obtained by evaluating the policy obtained from previous time step. For convenience, we denote policy $\hat{\pi}_t$ as the solution to maximizing the $L$-step lookahead planning objective, i.e.,

$$\hat{Q}_t(x(t), u_A(t), \hat{u}_H(t)) = \mathbf{E}[\sum_{i=0}^{L-1} \gamma^i r_A(\hat{x}(t+i), u_A(t+i), \hat{u}_H(t+i))$$
$$+ \gamma^L \hat{Q}_{t-1}(\hat{x}(t+L), u_A(t+L), \hat{u}_H(t+L))]$$
$$\hat{\pi}(x(t)|u_H) = \arg\max_{u_A(t)} \max_{\{u_A(t+1), \cdots, u_A(t+L-1)\}} \hat{Q}_t(x(t), u_A(t), \hat{u}_H(t)) \quad (2)$$

where $\hat{x}(t+i)$ is the state that the system will end up with if HV chose action $\hat{u}_H(t+i)$ and AV chose $u_A(t+i)$ at time step $t+i$. We denote $u_H = \{u_H(t)\}_{t=1}^{T}$. It can be seen that AV's policy is conditioned on HV's policy via $\hat{u}_H(t)$. We provide a detailed discussion on the relation to the model-free RL method and Actor-Critic in Appendix C.

**HV's Decision Making with Bounded Rationality.** HV's decision making has distinct characteristics. As mentioned by the pioneering study of behavior theory Simon (1957), individuals have constraints in both their *understanding of their surroundings* and their *computational capacities*. Additionally, they face search costs when seeking sophisticated information in order to devise optimal decision rules. Therefore, we propose to model human as responding to robots actions with bounded rationality. We additionally assume HV choose the action by planning for a short time horizon, in contrast to the long horizon planning in AV's decision making. Specifically, at time step $t$, HV chooses the (sub-optimal) action by planning ahead for $N$ steps, i.e.,

$$\Phi_H(x(t), u_A(t), u_H(t)) := \sum_{i=0}^{N-1} r_H(x(t+i), u_A(t+i), u_H(t+i)) \quad (3)$$

Meanwhile, to underscore the impact of the bounded rationality in HV's decision making, we use $u_H^*(t) := \arg\max_{u_H(t)} \max_{u_H(t+1), \cdots, u_H(t+N-1)} \Phi_H(x(t), u_A(t), u_H(t))$ to denote the optimal solution of Equation (3) and $u_H(t)$ to denote the sub-optimal action chosen by HV. Note that HV's policy is conditioned on AV's behavior $u_A(t)$ and we assume the time horizon $N$ is short enough such that the human can effectively extrapolate the robot's course of action, i.e., $u_A(t+i)$ is the true action taken by AV. We remark that we do not assume HV has access to the overall plan of AV but only the first few time steps. It has been shown in previous work Sadigh et al. (2018) that predicting a short-term sequence of controls is manageable for human, e.g., the AV will merge into HV's lane after a short period of time.

## 3 Characterization of HV and AV's Learning Performance

### 3.1 Regret of AV with $L$-step Lookahead Planning

In this subsection, we study the impact of bounded rationality and uncertainty-aware planning on the performance of AV. To this end, we first quantify the performance gap between choosing optimal actions and sub-optimal actions, for given HV's behavior *fixed*. Therefore, conditioned on HV's action $u_H = \{u_H(t)\}_{t=1}^{T}$, the regret for $T$ interaction of AV is defined as

$$\mathcal{R}_A(T|u_H) = \frac{1}{T}\sum_{t=1}^{T} \text{Reg}_A(t) := \mathbf{E}_{x \sim \rho_0}\left[\frac{1}{T}\sum_{t=1}^{T}\left(V^*(x|u_H(t)) - V^{\hat{\pi}_t}(x)\right)\right],$$

where we use $V^*(x|u_H(t))$ to denote the optimal value function attained by the optimal policy $\pi^*$ given HV's action $u_H$. $\hat{\pi}_t$ is the policy obtained in the $t$-th time step while AV solving $L$-step looka-head planning objective Equation (2) based on its prediction on HV's future actions. In particular, at each time step $t$, conditioned on HV's action $u_H(t)$, the optimal value function $V^*(x|u_H(t))$ is determined by choosing a policy $\pi_A^*(t)$ from policy space $\Pi_A$. Hence, the regret defined for AV is closely related to adaptive regret Loftin & Oliehoek (2022). Without loss of generality, we have a general model on HV's prediction error.

**AV's Prediction of HV's Actions.** Since HV's bounded rationality is unknown to AV and the accurate model on HV is thus challenging to obtain, we assume AV's prediction of HV's action $\hat{u}_H(t + l)$ has $\epsilon_A(t)$ difference from the HV's underlying real (sub-optimal) action $u_H(t + l)$, i.e.,

$$\hat{u}_H(t + l) = u_H(t + l) + \epsilon_A(t + l), \;\; l = 0, 1, 2, \cdots, L, \tag{4}$$

where $\epsilon_A(t) \sim \mathcal{N}(\mu_A, \sigma_A^2 I)$ is the AV's prediction error. Given the prediction on HV's actions, we first quantify the performance gap $\text{Reg}_A(t)$ of AV at each time-step $t$. Then we characterize the AV's learning performance in terms of regret $\mathcal{R}_A(T|u_H)$ in the non-linear case while considering the adaptive nature of AV's learning process, e.g., the time-varying function approximation error.

**An Illustrative Example: Performance Gap in the Linear Case.** For ease of exposition, we first consider the linear system dynamics model with system parameter $A, B_H, B_A$, i.e.,

$$x(t + 1) = Ax(t) + B_A u_A(t) + B_H u_H(t).$$

In the linear case, it is easy to see the resulting state transition model when AV is planning for the future steps based on the prediction of HV's action:

$$\hat{x}(t + l) = x(t + l) + \sum_{i=1}^{l} A^{i-1} B_H \epsilon_A(t + l - i), \;\; l = 1, 2, \cdots, \tag{5}$$

where we denote $x(t)$ as the real state when AV choose $u_A(t)$ and HV chooses $u_H(t)$. It can be seen that due to the *error accumulation* in AV's prediction, the state transition model tends to depart from the underlying true model significantly over prediction horizon $l$. Next, we present the performance for one interaction with assumptions on the function approximation error.

**Assumption 1.** The value function approximation error in the $t$-th step is $\epsilon_{v,t}(x) := V^*(x) - \hat{V}_t(x)$ with mean $\mathbf{E}_x[\epsilon_{v,t}(x)] = \mu_{v,t}$. The value function is upper bounded by $V_{\max}$.

In practice, the optimal value function can be estimated by using Monte-Carlo Tree Search (MCTS) over a class of policies or the offline training with expert prior Gelly & Silver (2011). Then, we have the following results on the performance gap in time-step $t$.

**Lemma 1** (AV's Performance Gap in the Linear Case.)**.** *Suppose Assumption 1 holds. Denote* $C_i = A^{i-1} B_H$. *Then we have the following upper bound on the performance gap of AV in the $t$-th step:*

$$\mathbf{E}\left[V^*(x|u_H) - V^{\hat{\pi}_t}(x)\right]$$

$$\leq \gamma^L \mu_{v,t} + \sum_{l=1}^{L} (V_{\max} + lR_{\max})\gamma^l \sqrt{\|\sum_{i=1}^{l} C_i \mu_A\|_2^2 + \|\sigma_A \left(\sum_{i=1}^{l} C_i C_i^\top\right)\|_F^2}.$$

**Error Accumulation in Planning.** In Lemma 1, we present a tight bound on the performance gap, where the first term in the upper bound is associated with the function approximation error and the second term is related to the AV's prediction error on HV's future action. Clearly, increasing the planning horizon $L$ can help to reduce the dependency on the accuracy of function approximation in a factor of $\gamma^L$ while risking the compounding error (the second term). Notably, the function approximation error $\mu_{v,t}$ will change during the learning process (ref. Equation (2)) and further have impact on AV's performance gap.

**Performance Gap in the Non-linear Case.** Observing the error accumulation in the linear case (ref. Equation 6), The disparity between the actual state and the predicted state, denoted as $x(t) - \hat{x}(t)$, tends to grow noticeably with time step $t$. Thus inspired, for the general case where the system model is unavailable, we formulate the prediction error as a diffusion process, i.e., denote $y(t) = x(t) - \hat{x}(t)$, then we have,

$$dy(t) = \mu_A dt + \Sigma_A^{1/2} dW(t), \; y(0) = 0,$$

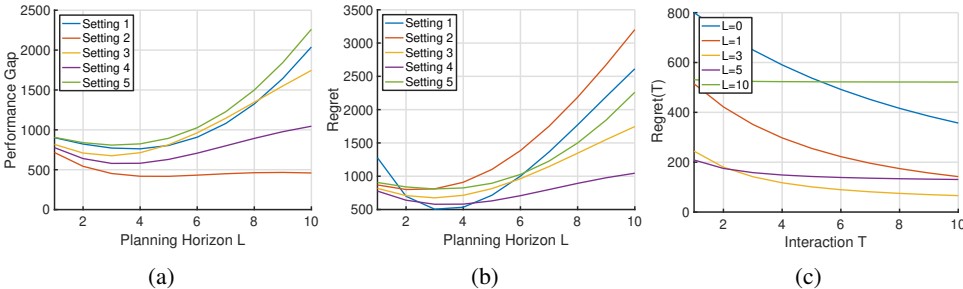

(a)            (b)            (c)

Figure 1: Numerical results on AV's regret. (a) The impact of planning horizon $L$ on AV's performance gap (ref. Lemma 2). (b) The impact of the planning horizon $L$ on AV's regret $\mathcal{R}_A$. (c) The impact of planning horizon on regret dynamics $\mathcal{R}_A(T)$ during the interactions.

where $t\mu_A$ is the drift term adn $t\Sigma_A := t\sigma_A^2 I$ is the variance term. $W(t)$ is the Weiner process. Then we can have the following results on the performance gap in the non-linear case.

**Lemma 2** (AV's Performance Gap in Non-linear Case). *Suppose Assumption 1 holds, then we have the upper bound of AV's performance gap in the $t$-th step as follows,*

$$\mathbf{E}\left[V^*(x|u_H) - V^{\hat{\pi}_t}(x)\right]$$

$$\leq \gamma^L \mu_{v,t} + \sum_{l=1}^L (V_{\max} + lR_{\max})\gamma^l \sqrt{\frac{(1+l)^2 l^2}{4}\|\mu_A\|_2^2 + \mathrm{tr}\left(\sigma_A^2 \frac{(1+l)l}{2}I\right)}.$$

**Goodhart's Law and Lookahead Length.** In Lemma 2, we examine the performance of AV through the lens of Goodhart's law, which predicts that increasing the optimization over a proxy beyond some critical point may degrade the performance on the true objective. In our case, the planning over predicted HV actions is equivalent to the optimization on a proxy object. Increasing the planning horizon is corresponding to increase the optimization pressure. As shown in Fig. 1a, where we plot the upper bound of the learning performance by changing different planning horizon $L$, the learning performance of AV clearly demonstrate the Goodhart's law, when increasing the planning horizon will initially help with the learning performance until a critical point. In practice, adjusting the look-ahead length (e.g., through grid search) is essential to enable AV to achieve the desired performance.

**Regret Analysis in the Non-linear Case.** To analyze the upper bound on the regret, we first impose the following standard assumptions on the MDP.

**Assumption 2** (Quadratic Reward Structure). The reward functions for AV and HV are the quadratic function of AV's action $u_A$ and HV's action $u_H$, respectively, i.e.,

$$r_H(x, u_A, u_H) = f_H(x, u_A) + u_H^\top S_H u_H$$
$$r_A(x, u_A, u_H) = f_A(x, u_H) + u_A^\top S_A u_A,$$

where $S_H$ and $S_A$ are positive definite matrix with largest eigenvalue $s_{\max}$. $f_H$ and $f_A$ are the reward functions that capture the influence of other agent and can be non-linear.

We note that Assumption 2 is commonly used in the analysis of regret especially in model-based RL Abbeel et al. (2006); Coates et al. (2008); Kolter et al. (2008) and the studies in mixed traffic Tian et al. (2022); Sadigh et al. (2016). In practice, the estimation of the parameter $S_H$ and $S_A$ can be achieved by various methods, e.g., Inverse Reinforcement Learning Tian et al. (2022). The limitations of Assumption 2 are discussed in Appendix D. Based on our findings in the performance gap, we now have the following result on the regret corresponding to AV's learning performance.

Let $C = \max_{u_A} u_A \mu_A^\top (\mu_A \mu_A^\top)^{-1}$ and $M$ be the cardinality of the action space $U_A$ and $U_H$. Denote $\lambda = \sqrt{\mathrm{eig}_{\max}(C^\top S_A C) s_{\max}}$. Then we have the following result.

**Theorem 3** (Regret on AV's Decision Making). *Suppose Assumptions 1 and 2 hold, the regret of AV's decision making over $T$ interactions is bounded above by*

$$\mathcal{R}_A(T) \leq \sum_{l=1}^{L}(V_{\max} + lR_{\max})\gamma^l\sqrt{\frac{(1+l)^2l^2}{4}\|\mu_A\|_2^2 + \text{tr}\left(\sigma_A^2\frac{(1+l)l}{2}I\right)}$$

$$+ \frac{\gamma^L}{T}\left(\Gamma\mu_{v,0} + \Lambda(s_{\max}M\sigma_A^2 + (s_{\max} + \lambda)\|\mu_A\|^2)\right),$$

*where $\Gamma := \frac{\gamma^{L+1}(1-\gamma^{T(L+1)})}{1-\gamma^{L+1}}$ and $\Lambda := \sum_{k=0}^{T}\prod_{i=0}^{k}\left(\gamma^{i(L+1)}\cdot\frac{\gamma(1-\gamma^L)}{1-\gamma}\right)$.*

**Reduce the Regret by Adjusting the Lookahead Length.** The upper bound in Theorem 3 is tight and it reveals the impact of the approximation error ($\mu_{v,0}$), prediction error ($\mu_A, \sigma_A$) and lookahead length $L$ on the learning performance. Specifically, we observe from the second term in the upper bound represents the *accumulation of the function approximation error*. The first term therein depends on the initial function approximation error $\mu_{v,0}$ and the last term is the compounding error due to the AV's prediction error during the $T$ times interactions. Our key observations are as follows: (1) Longer planning horizon, e.g., $L = 10$ in Fig. 1b and Fig. 1c, will likely make the prediction error more pronounced and dominate the upper bound. (2) While in the case when the planning horizon is short, e.g., $L = 1$ in Fig. 1b and Fig. 1c, we observe the function approximation error will likely dominate the upper bound. The empirical results provide the insighs on how to adjust the lookahead length in practice. For instance, if the function approximation error is more pronounced than the prediction error, it is beneficial of using longer planning horizon $L$. The proof of AV's regret is relegated to Appendix E.

## 3.2 REGRET OF HV WITH BOUNDED RATIONALITY

Given AV's action $u_A$, we define the regret for HV conditioned on AV's action $u_A$ as follows:

$$\mathcal{R}_H(T|u_A) = \mathbf{E}_{x(0)\sim\rho_0}\left[\frac{1}{T}\sum_{t=1}^{T}(\Phi_H^*(t) - \Phi(t))\right],$$

where $\Phi_H^*(t) := \Phi_H(x(t), u_H^*(t), u_A(t))$ is the optimal value and it is determined by choosing a policy $\pi_H^*(t)$ from policy space $\Pi_H$ such that $\Phi(x, \pi_A^t, \pi_H)$ is maximized. $\Phi(t) := \Phi_H(x(t), u_H(t), u_A(t))$ represents the value achieved when HV chooses sub-optimal action due to bounded rationality. For ease of exposition, we assume HV's decision making is myopic and HV's planning horizon is $N = 1$, such that $\Phi_H(x(t), u_A(t), u_H(t)) := r_H(x(t), u_A(t), u_H(t))$. Meanwhile, we assume HV makes sub-optimal decision as follows,

$$u_H(x(t), u_A(t)) = u_H^*(x(t), u_A(t)) + \epsilon_H(t)$$

where $\epsilon_H(t) \sim \mathcal{N}(\mu_H, \Sigma_H)$ is due to bounded rationality of humans and it is not known by AV.

Let $C_H = \max_{u_H} u_H\mu_H^\top(\mu_H\mu_H^\top)^{-1}$ and $\lambda_H = \sqrt{\text{eig}_{\max}(C_H^\top S_H C_H)s_{\max}}$, then we have the following results on the upper bound of HV's regret which shows the impact of bounded rationality on HV's performance. The proof of Theorem 4 is available in Appendix F.

**Theorem 4** (Regret for HV.). *Suppose Assumption 2 holds. Then we have the regret of HV's decision making over $T$ interactions to be bounded above by*

$$\mathcal{R}_H(T) \leq s_{\max}M\cdot\sigma_H^2 + (s_{\max} + \lambda_H)\|\mu_H\|^2$$

## 4 REGRET DYNAMICS IN MIXED AUTONOMY

Aiming to understand "*How do different decision making strategies impact the overall learning performance?*", especially on the impact of HV's bounded rationality on AV's performance, we study the regret dynamics in this section. More concretely, we denote the regret for the whole system as $\mathcal{R}_{A-H}(T)$, i.e., $\mathcal{R}_{A-H}(T) :=$

$$\frac{1}{T}\sum_{t=1}^{T}\left(\underbrace{\mathbf{E}\left[V^*(x|u_H^*(t)) - V^{\hat{\pi}_t}(x)\right]}_{(i)} + \underbrace{\mathbf{E}\left[\Phi(x(t), u_A^*(t), u_H^*(t)) - \Phi(x(t), u_A(t), u_H(t))\right]}_{(ii)}\right),$$

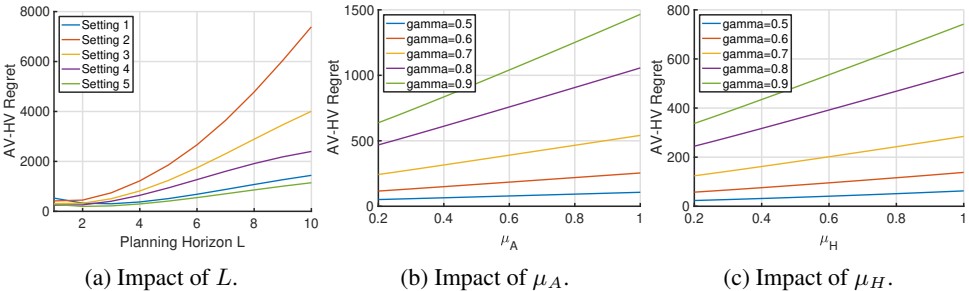

Figure 2: Empirical studies on AV and HV's decision making on the overall performance.

where $V^*(x|u_H^*(t))$ is the optimal value function when HV also takes the optimal action $u_H^*(t)$, e.g., $u_H^*(t) = \arg\max_{u_H} \Phi(x(t), u_A^*(t), u_H)$. Meanwhile $\Phi(x(t), u_A^*(t), u_H^*(t))$ is the optimal value when AV takes the optimal action $u_A^*(t) = \arg\max_{u_A} V^*(x, u_A, u_H^*(t))$ (without prediction error or function approximation error) while HV takes optimal action $u_H^*$. Intuitively, regret $\mathcal{R}_{A-H}(T)$ is defined as the difference between the best possible outcome, i.e., both AV and HV act and response to each other optimally, and the realized outcome, i.e., AV makes decision with prediction error and function approximation error while HV makes decisions with bounded rationality. Specifically, we note that the regret definition $\mathcal{R}_{A-H}$ can be naturally decomposed into two parts such that term (i) and term (ii) characterize the impact of HV's (AV's) decision making on AV (HV), respectively.

**Term (i).** Notice that term $(i)$ in $\mathcal{R}_{A-H}(T)$ can be decoupled as

$$V^*(x|u_H^*) - V^{\hat{\pi}_t}(x) := (V^*(x|u_H^*) - V^*(x|u_H)) + (V^*(x|u_H) - V^{\hat{\pi}_t}(x)).$$

The first term is induced by the sub-optimality of HV while the second term is the performance gap of AV, i.e., $\text{Reg}_A(t)$.

**Term (ii).** Similarly, we can decouple term $(i)$ into two parts,

$$\Phi(x(t), u_A^*(t), u_H(t)) - \Phi(x(t), u_A(t), u_H(t))$$
$$= \Phi(x(t), u_A^*(t), u_H^*) - \Phi(x(t), u_A(t), u_H^*) + \Phi(x(t), u_A(t), u_H^*) - \Phi(x(t), u_A(t), u_H(t)),$$

where the impact of AV's decision making is shown as the first term and the second term is the performance gap of HV, i.e., $\text{Reg}_H(t)$.

Denote $\Psi_A(l) = \sqrt{\frac{(1+l)^2 l^2}{4}\|\mu_A\|_2^2 + \text{tr}(\sigma_A^2 \frac{(1+l)l}{2} I)}$ and $\Psi_H(l) = \sqrt{\frac{(1+l)^2 l^2}{4}\|\mu_H\|_2^2 + \text{tr}(\sigma_H^2 \frac{(1+l)l}{2} I)}$. For ease of presentation, we use notation $\Psi_v = \Gamma\mu_{v,0} + \Lambda(s_{\max} M \sigma_A^2 + (s_{\max} + \lambda)\|\mu_A\|^2)$ to represent the regret term in Theorem 3 and $\Xi_H = s_{\max} M \cdot \sigma_H^2 + (s_{\max} + \lambda_H)\|\mu_H\|^2$ to represent the term in Theorem 4. Hence, building upon our results in Theorem 3 and Theorem 4, we give the upper bound of $\mathcal{R}_{A-H}(T)$ in the following corollary.

**Corollary 5** (Regret of the HV-AV Interaction System). *Suppose Assumptions 2 holds. Then we have the upper bound on the regret of AV-HV system as follows,*

$$\mathcal{R}_{A-H}(T) \leq \sum_{l=1}^{L} (V_{\max} + lR_{\max})\gamma^l (2\Psi_A(l) + \Psi_H(l)) + \Xi_H + \frac{1}{T}\gamma^L \Psi_v$$

Corollary 5 shows the impact of HV and AV's decision making on the overall learning performance through terms $\Psi_A, \Psi_v$ and $\Psi_H, \Xi_H$, respectively. In what follows, we conduct the empirically studies to thoroughly examine the impact of each agent while holding another agent fixed.

**Impact of AV's decision making on the overall system performance.** (1) Implications on choosing discounting factors. In Fig. 2b, we show the impact of the prediction error $\mu_A$ on the regret considering different discounting factor settings. Clearly, the larger discounting factor puts more emphasis on the future rewards comparing with the small discounting factor, which can "amplify" the impact of the prediction error $\mu_A$, e.g., the distance between each line for the same $\mu_A$. (2) Impact of function approximation error during interaction. In Fig. 3a, we study the impact of the function approximation error on the learning performance. As expected, the initial function approximation error $\mu_{v,0}$ have the huge impact on the regret in the first few interaction $T$. While during

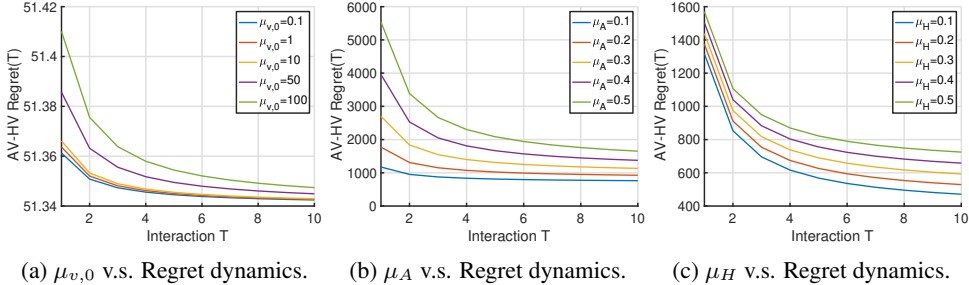

(a) $\mu_{v,0}$ v.s. Regret dynamics.    (b) $\mu_A$ v.s. Regret dynamics.    (c) $\mu_H$ v.s. Regret dynamics.

Figure 3: Empirical results on how AV and HV's decision making have impact on the overall regret dynamics, i.e., take regret as function of $T$.

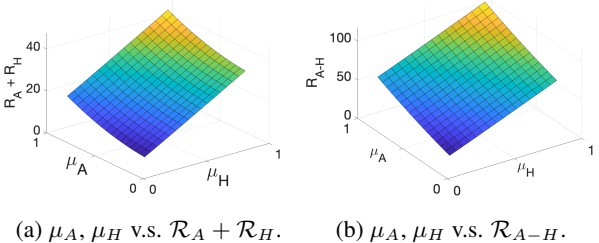

(a) $\mu_A$, $\mu_H$ v.s. $\mathcal{R}_A + \mathcal{R}_H$.    (b) $\mu_A$, $\mu_H$ v.s. $\mathcal{R}_{A-H}$.

Figure 4: Illustration of the impact of $\mu_A$, $\mu_H$ on the regret summation $\mathcal{R}_A + \mathcal{R}_H$ and the overall regret $\mathcal{R}_{A-H}$.

the learning process, the value function is updated and contributes less to the overall learning regret, e.g., the last term in Corollary 5. (3) Implications on the Priority of training: function approximation v.s. prediction model. In Fig. 3a and Fig. 3b we show the impact of different function approximation error ($\mu_{v,0}$) and prediction error ($\mu_A$) on the system regret. It can be seen that by reducing the prediction error from 0.4 to 0.2, the regret have significant change from 4000 to 1800 (-30%). While reducing the function approximation error from 100 to 50, the regret changes from 51.41 to 51.38 (-0.06%). The empirical results indicate that optimizting over prediction model tends to help us get more improvement on regret. The proof of Corollary 5 can be found in Appendix G.

**Impact of HV's Bounded Rationality on the overall system performance.** As illustrated in Fig. 2c, we conduct the experiments on the relationship between regret and human's decision making error $\mu_H$ by setting different discounting factors. In Fig. 3c, we can see that the regret difference caused by $\mu_H$ can be consistent during the interaction, which can be related to the second term in the upper bound of $\mathcal{R}_{A-H}$. Moreover, we also demonstrate the impact of HV's decision making on AV (and vice versa) in Fig. 4. For instance, in Figure 4b, a given $u_H$ will constrain the best possible outcome that AV can achieve, e.g., the projection on the $\mu_A$-Regret plane.

## 5    CONCLUSION AND FUTURE WORK

In this work, we take the regret analysis approach to address the questions 1) "*How does learning performance depend on HV's bounded rationality and AV's planning horizon?*" and 2) "*How do different decision making strategies between AV and HV impact the overall learning performance?*". To this end, we first propose a HV-AV interaction formulation which is able to capture the heterogeneous decision making of HV and AV. Based on the proposed formulation, we derive the upper bound on the regret for both HV and AV, respectively. By delving into the upper bound, we identify the Goodhart's law phenomenon in AV's decision making, where AV adopt the planning based RL using predicted human actions. Meanwhile we show the error accumulation effect in HV's decision making due to the bounded rationality in HV's decision making. Based on these results, we further analyze the impact of AV and HV's decision making on the overall system performance and we also derive the upper bound of the system regret. Through empirical study, we demonstrate how do different learning strategies affect the system performance. In this work, we assume that AV's prediction error is set to follow a fixed distribution. It is worth to explore the time varying prediction error distribution and further develop practical algorithms to address the regret minimization.

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

# Appendix

## A    EXTENSION TO MORE THAN TWO AGENTS

We remark that it is feasible to extend to more than one AV and one HV setting and we share some preliminary thoughts as follows. Assume there are $N_H$ HVs and $N_A$ AVs in the mixed traffic system. With abuse of notations, we define the action vector for AVs and HVs as follows, at time step $t$,

$$u_H(t) = [u_{H,1}(t), u_{H,2}(t), \cdots, u_{H,N_H}(t)]$$

$$u_A(t) = [u_{A,1}(t), u_{A,2}(t), \cdots, u_{A,N_A}(t)]$$

By defining the prediction error as in Equation (4) and HVs' bounded rationality as in Section 3.2, our analysis framework still can be applied. We remark that the dimenion of the approximation error term and the bounded rationality term is thus $N_A$ and $N_H$ times higher than the two-agent case. Hence, the resulting regret in Theorem 3 and Theorem 4 are $N_A$ and $N_H$ times higher than the two-agent case.

## B    DIFFERENCE IN PROBLEM SETTING

**Difference between our MDP formulation and Dec-POMDP.** We outline a few new ideas beyond the conventional Dec-POMDP as follows.

- In theoretical studies, many MARL formulations for Dec-POMDP, often assume all agents use the same RL algorithms to 'maximize' the rewards. Our work aims to study the case where AV and HV use different learning methods, i.e., longer-term look-ahead planning and myopic decision making to achieve their objectives.

- Further, in our setting, HV makes decision with bounded rationality at each time step, which deviates from reward maximization.

- (**Theoretical Results**) In our setting, we study the regret dynamics of the system (cf. $\mathcal{R}_{A-H}$ Section 4) such that the impact of different learning strategies on the system performance is characterized. Specifically, we show how HV's bounded rationality, AV's planning horizon and function approximation error have impact on the overall system dynamics. We remark this is different from the analysis in MARL formulation, where in general, the Nash Equilibrium is to be identified. Thus, the analysis method used in our work is very different from previous MARL formulation.

## C    GENERALIZATION OF AV AND HV'S LEARNING STRATEGIES.

**AV's Learning Strategies.** We clarify that Equation (2) can be degenerated into many commonly used RL algorithms, for instance,

- (Model-free Case) Set $L = 1$, Equation (2) is the model-free Q-function update and our regret analysis still holds.

- (Actor-Critic Case) Let $Q$-function and policy $\pi$ be parameterized by $\theta$ and $\phi$, respectively, Then Equation (2) can be learned by using Actor-Critic, i.e., in the actor step, $\theta$ is updated by maximizing the $L$-step look-ahead objective and $\phi$ is updated using policy gradient. Note that in this case, the approximation error in both Actor and Critic update can be encapsulate into $\epsilon_{v,t}$ as in Assumption 1. Our proof of the regret remains the same.

**HV's Learning Strategies.** In Equation (3), we consider AV's decision making to be $N$-step planning while we do not impose any constrains on the length of $N$. In particular, when $N \to \infty$, the decision making of HV is related to dynamic programming (assume the model is available) and otherwise, the decision making of AV is in the same spirit of Model Predictive Control (MPC).

# D LIMITATIONS OF ASSUMPTIONS

We summarize the limitations of two assumptions as follows:

- (Assumption 1: Function Approximation Error) In practice, since the underlying optimal value function is unknown, a commonly used approach to estimate the function approximation error $\epsilon_{v,t}$ and $\mu_{v,t}$ is to compare the difference between the rollout of the current policy (to estimate $\hat{V}_t$) and Monte-Carlo Tree Search (MCTS) (to estimate $V^*$) [R1]. However, in order to get an accurate estimation of the optimal value, MCTS need to try different policies and can be time-consuming if the state space and action space are large. One of the promising approach is to leverage an offline dataset with the interaction history between the HVs and AVs [R2].

- (Assumption 2: Reward Structure) In order to obtain the reward parameters $S_H$ and $S_A$ for HVs and AVs, various factors may be taken into considerations, e.g., safety, speed, comfort. In practice, the reward function design is an open question and highly depends on the problem of interest. For instance, [R3] also considers reference path deviation to avoid the vehicles to drive out of the lane. Handcraft all the factors that matter to the questions can be challenging. A promising way to efficiently learn such a reward signal can be achieved by Inverse Reinforcement Learning (IRL) based on HVs and AVs driving data.

# E PROOF OF AV'S REGRET.

**Proxy in the System Dynamics.** In the linear case, we first derive the resulting state transition model when AV is planning for the future steps while using the prediction of HV's action. The corresponding state dynamics can be written as, i.e., after observing $x(t)$,

$$
\begin{aligned}
\hat{x}(t+1) &= A\hat{x}(t) + B_A u_A(t) + B_H \hat{u}_H(t) \\
&= Ax(t) + B_A u_A(t) + B_H u_H(t) + B_H \epsilon_A(t) \\
&:= x(t+1) + B_H \epsilon_A(t)
\end{aligned}
$$

where $x(t+1)$ is the true state when AV and HV takes action $u_A(t)$ and $u_H(t)$.

Then at the next step, we have,

$$
\begin{aligned}
\hat{x}(t+2) &= A\hat{x}(t+1) + B_A u_A(t+1) + B_H \hat{u}_H(t+1) \\
&= A\hat{x}(t+1) + B_A u_A(t+1) + B_H \hat{u}_H(t+1) \\
&= Ax(t+1) + B_A u_A(t+1) + B_H u_H(t+1) + AB_H \epsilon_A(t) + B_H \epsilon_A(t+1)
\end{aligned}
$$

It can be seen that the estimated state and the real state has the following relationship,

$$
\hat{x}(t+l) = x(t+l) + \sum_{i=1}^{l} A^{i-1} B_H \epsilon_A(t+l-i). \tag{6}
$$

**Quantify the Regret.** Recall the definition of the regret (performance gap), i.e.,

$$
\text{Reg}_A(T) := \mathbf{E}_{x \sim \rho_0}\left[\frac{1}{T}\sum_{t=1}^{T}\left(V^*(x(t)) - V^{\hat{\pi}}(x(t)))\right]\right.
$$

$$
\text{Reg}_A(t) \triangleq \underbrace{V^*(x(t)) - V^{\pi_A}(x(t))}_{\text{FA Error}} + \underbrace{V^{\pi_A}(x(t)) - V^{\hat{\pi}_A}(x(t))}_{\text{Modeling Error and Lookahead}}
$$

$$
:= \underbrace{V^*(x(t)) - V^{\pi}(x(t))}_{(1)} + \underbrace{V^{\pi}(x(t)) - V^{\hat{\pi}}(x(t))}_{(2)} \tag{7}
$$

For simplicity, we define the following notations,

$\hat{\tau}$ trajectory obtained by running $\hat{\pi}_A$ with function approximation error (FA)

$\tau$ trajectory obtained by running $\pi$ with FA error

$\tau^*$ trajectory obtained by running in $M$ without FA error

$u_t = (u_A(t), u_H(t))$

Meanwhile, we use $\hat{\pi}$ to denote the policy obtained by running lookahead on a inaccurate model and $\pi$ is the policy using the accurate model. Note that in both cases, the terminal cost are estimated by $\hat{V}$ (with function approximation error).

**Part 1. Impact of the Function Approximation Error.** We first quantify the first term (1) in Equation (7) as follows,

$$
\begin{aligned}
V^*(x_0) - V^\pi(x_0) =& \mathbb{E}_{\tau^*}\left[\sum \gamma^t r(x_t, u_t) + \gamma^L V^*(s_L)\right] - \mathbb{E}_\tau\left[\sum \gamma^t r(x_t, u_t) + \gamma^L V^\pi(x_L)\right]\\
=& \mathbb{E}_{\tau^*}\left[\sum \gamma^t r(x_t, u_t) + \gamma^L V^*(x_L)\right] - \mathbb{E}_\tau\left[\sum \gamma^t r(x_t, u_t) + \gamma^L V^*(x_L)\right]\\
&+ \mathbb{E}_\tau\left[\sum \gamma^t r(x_t, u_t) + \gamma^L V^*(x_L)\right] - \mathbb{E}_\tau\left[\sum \gamma^t r(x_t, u_t) + \gamma^L V^\pi(x_L)\right]\\
=& \mathbb{E}_{\tau^*}\left[\sum \gamma^t r(x_t, u_t) + \gamma^L V^*(x_L)\right] - \mathbb{E}_\tau\left[\sum \gamma^t r(x_t, u_t) + \gamma^L V^*(x_L)\right]\\
&+ \gamma^L \mathbb{E}_\tau\left[V^*(x_L) - V^\pi(x_L)\right]
\end{aligned}
\tag{8}
$$

Assumptions on the approximation error. We assume that the function approximation error is $\epsilon_v$ with mean $\mu_v$ and variance $\Sigma_v$, i.e.,

$$
V^*(x) - \hat{V}(x) = \epsilon_v(x)
$$

Bring the above relation to the first two terms of Equation (8) gives us,

$$
\mathbb{E}_{\tau^*}\left[\sum \gamma^t r(x_t, u_t) + \gamma^L V^*(x_L)\right] = \mathbb{E}_{\tau^*}\left[\sum \gamma^t r(x_t, u_t) + \gamma^L \hat{V}(x)(x_L) + \gamma^L \epsilon_v(x_L)\right]
$$

$$
\mathbb{E}_\tau\left[\sum \gamma^t r(x_t, u_t) + \gamma^L V^*(x_L)\right] = \mathbb{E}_\tau\left[\sum \gamma^t r(x_t, u_t) + \gamma^L \hat{V}(x)(x_L) + \gamma^L \epsilon_v(x_L)\right]
$$

Then we have,

$$
\begin{aligned}
&V^*(x_0) - V^\pi(x_0)\\
&= \mathbb{E}_{\tau^*}\left[\sum \gamma^t r(x_t, u_t) + \gamma^L \hat{V}(x)(x_L) + \gamma^L \epsilon_v(x_L)\right]\\
&\quad - \mathbb{E}_\tau\left[\sum \gamma^t r(x_t, u_t) + \gamma^L \hat{V}(x)(x_L) + \gamma^L \epsilon_v(x_L)\right]\\
&\quad + \gamma^L \mathbb{E}_\tau\left[V^*(x_L) - V^\pi(x_L)\right]\\
&= \mathbb{E}_{\tau^*}\left[\sum \gamma^t r(x_t, u_t) + \gamma^L \hat{V}(x)(x_L)\right]\\
&\quad - \mathbb{E}_\tau\left[\sum \gamma^t r(x_t, u_t) + \gamma^L \hat{V}(x)(x_L)\right] + \gamma^L\left(\mathbb{E}_{\tau^*}[\epsilon_v(x_L)] - \mathbb{E}_\tau[\epsilon_v(x_L)]\right)\\
&\quad + \gamma^L \mathbb{E}_\tau\left[\hat{V}(x_L) + \epsilon_v(x_L) - V^\pi(x_L)\right]\\
&= \mathbb{E}_{\tau^*}\left[\sum \gamma^t r(x_t, u_t) + \gamma^L \hat{V}(x)(x_L)\right] - \mathbb{E}_\tau\left[\sum \gamma^t r(x_t, u_t)\right]\\
&\quad + \gamma^L \mathbb{E}_{\tau^*}[\epsilon_v(x_L)] - \gamma^L \mathbb{E}_\tau[V^\pi(x_L)]\\
&= \left(\mathbb{E}_{\tau^*}\left[\sum \gamma^t r(x_t, u_t)\right] - \mathbb{E}_\tau\left[\sum \gamma^t r(x_t, u_t)\right]\right)\\
&\quad + \gamma^L\left(\mathbb{E}_{\tau^*}\left[\hat{V}(x)(x_L)\right] - \mathbb{E}_\tau\left[\hat{V}(x)(x_L)\right]\right) + \gamma^L \mathbb{E}_{\tau^*}[\epsilon_v(x_L)]
\end{aligned}
$$

First term: (1) $\left(\mathbb{E}_{\tau^*}\left[\sum \gamma^t r(x_t, u_t)\right] - \mathbb{E}_\tau\left[\sum \gamma^t r(x_t, u_t)\right]\right)$.

Assume the reward function is bounded by $R_{\min} \leq r(x, u) \leq R_{\max}, \forall (x, u)$. Then we have

$$
\frac{1-\gamma^L}{1-\gamma}(R_{\min} - R_{\max}) \leq (1) \leq \frac{1-\gamma^L}{1-\gamma}R_{\max}
$$

Second term: (2) $\gamma^L\left(\mathbb{E}_{\tau^*}\left[\hat{V}(x)(x_L)\right] - \mathbb{E}_\tau\left[\hat{V}(x)(x_L)\right]\right)$. By assuming the function approximation value is bounded by $[\hat{V}_{\min}, \hat{V}_{\max}]$, we have,

$$
\gamma^L(\hat{V}_{\min} - \hat{V}_{\max}) \leq (2) \leq \gamma^L \hat{V}_{\max}
$$

Second term: (3) $\gamma^L \mathbb{E}_{\tau^*}[\epsilon_v(x_L)]$

$$\gamma^L \epsilon_{v,\min} \leq (3) \leq \gamma^L \epsilon_{v,\max}$$

Alternatively we have

$$(3) = \gamma^L \mu_v$$

By combing all three parts, we have the upper bound and lower bound as follows,

$$V^*(x_0) - V^\pi(x_0) \leq \frac{1 - \gamma^L}{1 - \gamma} R_{\max} + \gamma^L \hat{V}_{\max} + \gamma^L \epsilon_{v,\max}$$

$$V^*(x_0) - V^\pi(x_0) \geq \frac{1 - \gamma^L}{1 - \gamma}(R_{\min} - R_{\max}) + \gamma^L(\hat{V}_{\min} - \hat{V}_{\max}) + \gamma^L \epsilon_{v,\min}$$

**Part 2. The Impact of the Modeling Error in the $L$-step Planning.** Now we are ready to quantify the second term in Equation (7).

We first define $U_l$ as follows. For any $0 \leq l \leq L$, define $U_l$ to be the $l$-step value expansion that rolls out the true model $P$ for the first $l$ steps and the approximate model $\hat{P}$ for the remaining $L - l$ steps:

$$U_l = \sum_{t=0}^{l-1} \gamma^t \mathbb{E}_{x_t \sim P_t^\pi(\cdot|x)}[R^\pi(x_t)] + \sum_{t=l}^{L-1} \gamma^t \mathbb{E}_{x_t \sim \hat{P}_{t-l}^\pi \circ P_l^\pi(\cdot|x)}[R^\pi(x_t)]$$
$$+ \gamma^L \mathbb{E}_{x_L \sim \hat{P}_{L-l}^\pi \circ P_l^\pi(\cdot|x)}\left[\hat{V}(x_L)\right],$$

where $\hat{P}_{L-l}^\pi \circ P_l^\pi(\cdot \mid x)$ denotes the distribution over states after rolling out $l$ steps with $P$ and $t - l$ steps with $\hat{P}$.

$$\hat{P}_{t-l}^\pi \circ P_l^\pi(\cdot \mid x) = \sum_{x' \in \mathcal{X}} P_l^\pi(x' \mid x) \hat{P}_{t-l}^\pi(\cdot \mid x')$$

Then we have,

$$U_L = V^\pi(x(t))$$
$$U_0 = V^{\hat{\pi}}(x(t))$$

Hence we have,

$$V^\pi(x(t)) - V^{\hat{\pi}}(x(t)) = U_L - U_0 = \sum_{l=0}^{L-1} U_{l+1} - U_l$$

To analyze each term in the sum, we re-arrange $U_l$ in the following ways

$$U_l = \sum_{t=0}^{l-1} \gamma^t \mathbb{E}_{x_t \sim P_t^\pi(\cdot|x)}[R^\pi(x_t)] + \gamma^l \mathbb{E}_{x_l \sim P_l^\pi(\cdot|x)}\left[V_{L-l}^{\hat{\pi}}(x_l)\right] \tag{9}$$

$$U_l = \sum_{t=0}^{l} \gamma^t \mathbb{E}_{x_t \sim P_t^\pi(\cdot|x)}[R^\pi(x_t)] + \gamma^{l+1} \mathbb{E}_{x_{l+1} \sim \hat{P}^\pi \circ P_l^\pi(\cdot|x)}\left[V_{L-l-1}^{\hat{\pi}}(x_{l+1})\right]. \tag{10}$$

where we denote $V_L^{\hat{\pi}}(x_l) := \sum_{t=0}^{L-1} \gamma^t \mathbb{E}_{x_t \sim \hat{P}_t^\pi(x)}[R^\pi(x_t)] + \gamma^L \mathbb{E}_{x_L \sim \hat{P}_L^\pi(x)}\left[\hat{V}(x_H)\right]$. Note that $\hat{V}$ is not the same as $V^{\hat{\pi}}$, where the latter represents the value of running the current policy $\hat{\pi}$ with $L$ step lookahead planning over a inaccurate model with a terminal cost estimation $\hat{V}$.

Now applying Equation (10) to $U_l$ and Equation (9) to $U_{l+1}$, then we have,

$$U_{l+1} - U_l = \sum_{t=0}^{l} \gamma^t \mathbb{E}_{x_t \sim P_t^\pi(\cdot|x)} [R^\pi(x_t)] + \gamma^{l+1} \mathbb{E}_{x_{l+1} \sim P_{l+1}^\pi(\cdot|x)} \left[ V_{\hat{P},L-l-1}^\pi(x_{l+1}) \right]$$

$$- \sum_{t=0}^{l} \gamma^t \mathbb{E}_{x_t \sim P_t^\pi(\cdot|x)} [R^\pi(x_t)] - \gamma^{l+1} \mathbb{E}_{x_{l+1} \sim \hat{P}^\pi \circ P_l^\pi(\cdot|x)} \left[ V_{\hat{P},L-l-1}^\pi(x_{l+1}) \right]$$

$$= \gamma^{l+1} \mathbb{E}_{x_l \sim P_l^\pi(\cdot|x), u_l \sim \pi(\cdot|x_l)} \left[ \mathbb{E}_{x' \sim P(\cdot|x_l,u_l)} \left[ V_{L-l-1}^{\hat{\pi}}(x') \right] \right.$$

$$\left. - \mathbb{E}_{x' \sim \hat{P}(\cdot|x_l,u_l)} \left[ V_{L-l-1}^{\hat{\pi}}(x') \right] \right]$$

$$= \gamma^{l+1} \mathbb{E}_{x_l \sim P_l^\pi(\cdot|x), u_l \sim \pi(\cdot|x_l)} \left[ \int_{x'} \left( P(x' \mid x_l, u_l) - \hat{P}(x' \mid x_l, u_l) \right) V_{L-l-1}^{\hat{\pi}}(x') dx' \right]$$

$$:= \gamma^{l+1} \mathbb{E}_{x_l \sim P_l^\pi(\cdot|x), u_l \sim \pi(\cdot|x_l)} [D(x_{l+1}|P, \hat{P})],$$

where we denote $D(x_{l+1}|P, \hat{P}) = \int_{x'} \left( P(x' \mid x_l, u_l) - \hat{P}(x' \mid x_l, u_l) \right) V_{L-l-1}^{\hat{\pi}}(x') dx'$.

It can be seen that $D(x_{l+1})$ is directly relevant to the lookahead length $l$ and the modeling error $\hat{P} - P$. In the linear case, the longer lookahead length makes the difference between $P$ and $\hat{P}$ more significant, i.e., Equation (6). Next, we give the expression for $D(x_{l+1}|P, \hat{P})$ to show its relation with the lookahead length $L$.

$$D(x_{l+1}|P, \hat{P}) = \int_{x'} \left( P(x' \mid x_l, u_l) - \hat{P}(x' \mid x_l, u_l) \right) V_{L-l-1}^{\hat{\pi}}(x') dx'$$

Linear Case. Recall Equation (6),

$$\hat{x}(t+l) = x(t+l) + \sum_{i=1}^{l} A^{i-1} B_H \epsilon_A(t+l-i),$$

where $\epsilon_A \sim \mathcal{N}(\mu_A, \Sigma_A)$. Then we have,

$$\hat{P}(x' \mid x_l, u_l) = \mathbb{P}(\sum_{i=1}^{l} A^{i-1} B_H \epsilon_A(t+l-i) = x' - Ax_l - Bu_l)$$

Given $\epsilon_A$ follows Gaussian distribution, we have

$$\sum_{i=1}^{l} A^{i-1} B_H \epsilon_A(t+l-i) \sim \mathcal{N}(\sum_{i=1}^{l} A^{i-1} B_H \mu_A, \sum_{i=1}^{l} A^{i-1} B_H \Sigma_A (A^{i-1} B_H)^\top)$$

Then we have

$$\sum_{i=1}^{l} A^{i-1} B_H \epsilon_A(t+l-i) \sim \mathcal{N}(\sum_{i=1}^{l} C_i \mu_A, \sigma_A^2 \sum_{i=1}^{l} C_i C_i^\top)$$

where $C_i := A^{i-1} B_H$.

For simplicity, assume $A^{i-1} B_H = I$, then we have

$$\sum_{i=1}^{l} A^{i-1} B_H \epsilon_A(t+l-i) \sim \mathcal{N}(l \cdot \mu_A, l\sigma_A^2 I)$$

Meanwhile, we have the underlying true dynamics of the system is

$$x(t+1) = Ax(t) + B_A u_A(t) + B_H u_H(t) + \epsilon_p(t).$$

Then we have,

$$P\left(x' \mid x_l, u_l\right) = \mathbb{P}(\epsilon_p = x' - Ax_l - Bu_l)$$

Notice that $\epsilon_p \sim \mathcal{N}(0, \sigma_p^2 I)$.

Then the difference between $P$ and $\hat{P}$ boils down to the difference between two Normal distribution. We have the following results,

$$W(\hat{P}, P) = \sqrt{\|\sum_{i=1}^{L} C_i \mu_A\|_2^2 + \|(\sigma_A \left(\sum_{i=1}^{l} C_i C_i^\top\right) - \sigma_p)I\|_F^2}$$

Or in the simple case

$$W(\hat{P}, P) = \sqrt{l^2 \|\mu_A\|_2^2 + \|(\sigma_A \sqrt{l} - \sigma_p)I\|_F^2}$$

Assume the value function is bounded by $V_{\max} = \sup_h \left\|\hat{V}_l^{\hat{\pi}}\right\|_L$, i.e., the maximum Lipschitzness of the estimated value function over all possible horizons. Now we have,

$$U_{l+1} - U_l \le V_{\max} \gamma^{l+1} \mathbb{E}_{x_{l+1}}[D(x_{l+1})] \le V_{\max} \gamma^{l+1} \mathbb{E}_{x_{l+1}}[W(\hat{P}, P)] \tag{11}$$

where $W$ is the Wasserstein distance.

Then we have

$$U_L - U_0 \le V_{\max} \sum_{l=1}^{L} \gamma^l \mathbb{E}_{x_{l+1}, u_{l+1} \sim \pi}[W(\hat{P}(\cdot|x, u), P(\cdot|x, u))]$$

**Combing two parts gives upper bound.**

By adding the upper bound of the two parts, we obtain the upper bounds and lower bound for the performance difference,

Linear Case, no FA error. In this case, we have the regret as follows,

$$\text{Reg}_A(t) \le \frac{1 - \gamma^L}{1 - \gamma} R_{\max} + V_{\max} \sum_{l=1}^{L} \sqrt{l^2 \|\mu_A\|_2^2 + \|(\sigma_A \sqrt{l} - \sigma_p)I\|_F^2}$$

Linear Case, with FA error.

$$\text{Reg}_A(t) \le \frac{1 - \gamma^L}{1 - \gamma} R_{\max} + V_{\max} \sum_{l=1}^{L} \sqrt{l^2 \|\mu_A\|_2^2 + \|(\sigma_A \sqrt{l} - \sigma_p)I\|_F^2} + \gamma^L \epsilon_{v,\max}$$

Non-linear Case, with FA error.

$$\text{Reg}_A(t) \le \frac{1 - \gamma^L}{1 - \gamma} R_{\max} + \gamma^L \hat{V}_{\max} + \gamma^L \epsilon_{v,\max}$$
$$+ V_{\max} \sum_{l=1}^{L} \gamma^l \mathbb{E}_{x_{l+1}, u_{l+1} \sim \pi}[W(\hat{P}(\cdot|x, u), P(\cdot|x, u))]$$

**Treat the Prediction Error as a Diffusion Process.** Recall the diffusion process:

$$dx(t) = \mu dt + \sigma dW(t)$$
$$\text{Drift: } \mu t = \mathbb{E}[x(t) - x(0)]$$
$$\text{Variance: } \sigma^2 t = \text{Var}[x(t) - x(0)]$$

where $W(t)$ is a wiener process, i.e., $dW(t) = \varepsilon_t \sqrt{dt}$, $\varepsilon_t \sim \mathcal{N}(0,1)$. Alternatively in the discrete case, we have $x(t) - x(0) = \mu t + \sigma W(t)$. In our setting, due to the compounding error in the lookahead planning, the difference between true state and predicted state becomes more and more different as the time horizon expands. Define the difference between the true state and predicted state as $y(t) = \hat{x}(t) - x(t)$, then we assume the prediction error follows a diffusion process, i.e.,

$$dy(t) = \mu_A dt + \Sigma_A dW(t), \ y(0) = 0$$

For simplicity, assume $\Sigma_A = \sigma_A^2 I$.

Then we can obtain that at time $t$, the prediction error follows a Gaussian distribution, i.e., $y(t) \sim \mathcal{N}(t\mu_A, t\sigma_A^2 I)$. Then we have the Wasserstein distance $\hat{P}$ and $P$ as follows Delon & Desolneux (2020),

$$W(\hat{P}_{l+1} - P) = \sqrt{\frac{(1+l)^2 l^2}{4} \|\mu_A\|_2^2 + \text{tr}\left(\sigma_A^2 \frac{(1+l)l}{2} I + \sigma_p^2 I - 2\sigma_A^2 \sigma_p^2 \frac{(1+l)l}{2} I\right)}$$

Finally, we obtain the upper bound for the non-linear case as follows:

$$\text{Reg}_A(t) \leq \frac{1-\gamma^L}{1-\gamma} R_{\max} + \gamma^L \hat{V}_{\max} + \gamma^L \mu_{v,t}$$
$$+ V_{\max} \sum_{l=1}^{L} \gamma^l \sqrt{\frac{(1+l)^2 l^2}{4} \|\mu_A\|_2^2 + \text{tr}\left(\sigma_A^2 \frac{(1+l)l}{2} I + \sigma_p^2 I - 2\sigma_A^2 \sigma_p^2 \frac{(1+l)l}{2} I\right)}$$

**Regret over time $T$.** Now we consider the regret over time $t = 1, 2, \cdots, T$. Assume the current policy is $\hat{\pi}_t$ and the learned value function is $\hat{V}_t$. Recall that AV chose its policy in the following way,

- Estimate value function using policy $\hat{\pi}_t$:

$$\hat{Q}_{t+1} = \left(\sum_{i=1}^{L} \mathbb{E}\left[\gamma^i r_A(\hat{x}(t+i), u_A(t+i), \hat{u}_H(t+i))\right]\right.$$
$$\left. + \gamma^{L+1} \hat{Q}_t(\hat{x}(t+L+1), \hat{u}(t+L+1))\right),$$

- Derive the greedy policy (as in MPC):

$$\hat{\pi}_{t+1} = \arg \max_{u_A(t+1)} \max_{u_A(t+2), \cdots, u_A(t+L)} \hat{Q}_{t+1}$$

It can be seen that due to the update of the value function $\hat{Q}$. Next we show the difference between $\hat{Q}_{t+1}$ and $\hat{Q}_t$. Recall that we assume $V^* - \hat{V}_t = \epsilon_v$, and we denote (with abuse of notation) $Q^* - \hat{Q}_t = \epsilon_t$. Now we have

$$\hat{Q}_{t+1} - Q^* = \gamma^{L+1} \epsilon_t + \sum_{i=1}^{L} \gamma^i (\hat{r}_A - r_A),$$

where we denote $\hat{r}_A = r_A(\hat{x}(t+i), u_A(t+i), \hat{u}_H(t+i))$ and $r_A = r_A(\hat{x}(t+i), u_A(t+i), u_H(t+i))$. Similar to the analysis to HV regret, we have

$$\mu_{v,t+1} := \mathbb{E}[\epsilon_{t+1}] \leq \gamma^{L+1} \mu_{v,t} + \frac{\gamma(1-\gamma^L)}{1-\gamma}(\text{Reg}_A)$$

where $\text{Reg}_A = ms\sigma_A^2 + (s + \lambda)\|\mu_A\|^2$.

Now we are ready to derive the regret for AV as follows,

$$
\begin{aligned}
\text{Reg}_A(T) =& \frac{1}{T}\sum_{t=1}^{T}\text{Reg}(t) \\
\leq & \left(\frac{1-\gamma^L}{1-\gamma}R_{\max} + \gamma^L\hat{V}_{\max}\right. \\
& \left. + V_{\max}\sum_{l=1}^{L}\gamma^l\sqrt{\frac{(1+l)^2l^2}{4}\|\mu_A\|_2^2 + \text{tr}\left(\sigma_A^2\frac{(1+l)l}{2}I + \sigma_p^2 I - 2\sigma_A^2\sigma_p^2\frac{(1+l)l}{2}I\right)}\right) \\
& + \frac{\gamma^L}{T}\left(\frac{\gamma^{L+1}(1-\gamma^{T(L+1)})}{1-\gamma^{L+1}}\mu_{v,0} + \sum_{k=0}^{T}\prod_{i=0}^{k}\left(\gamma^{i(L+1)}\cdot\frac{\gamma(1-\gamma^L)}{1-\gamma}\text{Reg}_A\right)\right) \\
=& \sum_{l=1}^{L}(V_{\max} + lR_{\max})\gamma^l\sqrt{\frac{(1+l)^2l^2}{4}\|\mu_A\|_2^2 + \text{tr}\left(\sigma_A^2\frac{(1+l)l}{2}I\right)} \\
& + \frac{\gamma^L}{T}\left(\Gamma\mu_{v,0} + \Lambda(s_{\max}M\sigma_A^2 + (s_{\max}+\lambda)\|\mu_A\|^2)\right),
\end{aligned}
$$

where $\Gamma := \frac{\gamma^{L+1}(1-\gamma^{T(L+1)})}{1-\gamma^{L+1}}$ and $\Lambda := \sum_{k=0}^{T}\prod_{i=0}^{k}\left(\gamma^{i(L+1)}\cdot\frac{\gamma(1-\gamma^L)}{1-\gamma}\right)$.

## F  PROOF OF HV'S REGRET.

Due to the bounded rationality, HV does not choose the optimal action and thus introduces the regret as follows

$$
\begin{aligned}
\text{Reg}_H(T) :=& \frac{1}{T}\sum_{t=1}^{T} \\
\text{Reg}_H(t) =& \mathbb{E}\left[\Phi(x(t), u_H^*(t), \hat{u}_A(t)) - \Phi(x(t), u_H(t), \hat{u}_A(t))\right],
\end{aligned}
$$

where we assume that HV can observe the action of AV in a timely manner. Next, we impose the assumptions on the reward structure to be quadratic, i.e.,

$$
r_H(x, u_A, u_H) = f_H(x, u_A) + u_H^\top S_H u_H, \tag{12}
$$

where $S_H$ are positive definite matrices.

Then we have the regret for HV to be,

$$
\begin{aligned}
\text{Reg}_H(t) =& \mathbb{E}\left[(u_H^*(t))^\top S_H u_H^*(t) - (u_H(t))^\top S_H u_H(t)\right] \\
=& \frac{1}{2}\mathbb{E}\left[(u_H^*(t) + u_H(t))^\top S_H (u_H^*(t) - u_H(t))\right. \\
& \left. + (u_H^*(t) - u_H(t))^\top S_H (u_H^*(t) + u_H(t))\right] \\
=& \frac{1}{2}\mathbb{E}\left[\left((u_H^*(t) + u_H(t))^\top S_H\epsilon_H(t) + \epsilon_H(t)^\top S_H (u_H^*(t) + u_H(t))\right)\right] \\
=& \frac{1}{2}\mathbb{E}\left[\left((2u_H^*(t) + \epsilon_H(t))^\top S_H\epsilon_H(t) + \epsilon_H(t)^\top S_H (2u_H^*(t) + \epsilon_H(t))\right)\right] \\
=& \mathbb{E}\left[\epsilon_H(t)^\top S_H\epsilon_H(t) + u_H^*(t)^\top S_H\epsilon_H(t) + \epsilon_H(t)^\top S_H u_H^*(t)\right] \\
=& \text{Tr}(S_H\Sigma_H) + \mu_H^\top S_H\mu_H + u_H^*(t)^\top S_H\mu_H + \mu_H^\top S_H u_H^*(t)
\end{aligned}
$$

Furthermore, we have the following assumptions on the matrices

- $\Sigma_H = \sigma_H I$, where $I$ is an identity matrix.
- The dimension of the action space is $n$
- $0 < s_{\min} \leq \text{eig}(S_H) \leq s_{\max}$, where $\text{eig}(S_H)$ is the eigenvalue of $S_H$.
- There exist a matrix $C$ such that $C_{\min}\mu_H \leq u_H^*(t) \leq C\mu_H$, notice that $u_H^*$ depends on AV's action.

With those assumptions in place, we have the upper bound for the regret as follows:

$$\text{Reg}_H(T) \leq n s_{\max} \cdot \sigma_H^2 + (s_{\max} + \lambda)\|\mu_H\|^2$$

where $\lambda := \sqrt{\text{eig}_{\max}(C^\top S_H C) \cdot s_{\max}}$.

## G  PROOF OF COROLLARY 5

We denote the regret for the whole system as $\mathcal{R}_{A-H}(T)$, i.e., $\mathcal{R}_{A-H}(T) :=$

$$\frac{1}{T}\sum_{t=1}^{T}\left(\underbrace{\mathbf{E}\left[V^*(x|u_H^*(t)) - V^{\hat{\pi}_t}(x)\right]}_{(i)} + \underbrace{\mathbf{E}\left[\Phi(x(t), u_A^*(t), u_H^*(t)) - \Phi(x(t), u_A(t), u_H(t))\right]}_{(ii)}\right),$$

where $V^*(x|u_H^*(t))$ is the optimal value function when HV also takes the optimal action $u_H^*(t)$, e.g., $u_H^*(t) = \arg\max_{u_H} \Phi(x(t), u_A^*(t), u_H)$. Notice that both term (i) and term (ii) can be decomposed in the following way

$$V^*(x|u_H^*(t)) - V^{\hat{\pi}_t}(x)$$
$$= \underbrace{V^*(x|u_H^*(t)) - V^*(x|u_H(t))}_{(a)} + \underbrace{V^*(x|u_H(t)) - V^{\hat{\pi}_t}(x)}_{(b)}$$

$$\Phi(x(t), u_A^*(t), u_H^*(t)) - \Phi(x(t), u_A(t), u_H(t))$$
$$= \underbrace{\Phi(x(t), u_A^*(t), u_H^*(t)) - \Phi(x(t), u_A(t), u_H^*(t))}_{(a)} + \underbrace{\Phi(x(t), u_A(t), u_H^*(t)) - \Phi(x(t), u_A(t), u_H(t))}_{(b)}$$

In the decomposition above, term (b) is related to AV and HV's regret, respectively. Now we quantify term (a).

**AV.** Term (a) is related to $V^*$ function and we need to show that due to the bounded rationality of HV, it has direct impact on AV's overall best possible performance, i.e., denote the trajectory collected by running through MDP $M$ with HV's action $u_H^*$ as $\tau^{\text{opt}}$, while the trajectory collected with HV's action $u_H$ is denoted as $\tau$, then we have

$$(a) = \mathbb{E}_{\tau^{\text{opt}}}\left[\sum\gamma^t r(x_t, u_t) + \gamma^L V^*(x_L)\right] - \mathbb{E}_\tau\left[\sum\gamma^t r(x_t, u_t) + \gamma^L V^*(x_L)\right]$$

$$= \mathbb{E}_{\tau^{\text{opt}}}\left[\sum\gamma^t r(x_t, u_t)\right] - \mathbb{E}_\tau\left[\sum\gamma^t r(x_t, u_t)\right] + \mathbb{E}_{\tau^{\text{opt}}}\left[\gamma^L V^*(s_L)\right] - \mathbb{E}_\tau\left[\gamma^L V^*(s_L)\right]$$

$$= \sum_{i=1}^{L}\gamma^i\left(\eta^{i,\text{opt}}(x, u) - \eta^i(x, u)\right)r(x, u) + \gamma^L\int_x \mathbb{P}[x|s_{L-1}, u_{L-1}^*] - \mathbb{P}[x|x_{L-1}, u_{L-1}]V^*(x)$$

$$\leq \sum_{i=1}^{L}\gamma^i \cdot i\epsilon_m r_{\max} + \gamma^L L V_{\max}\epsilon_m,$$

where $\epsilon_m$ is the total variation between $M$ and $\hat{M}$ due to HV's noisy action as the disturbance is upper bounded by $\epsilon_m$. The explicate formulation of the upper bound is available in Prop. 2.1 Devroye et al. (2018).

**HV.** Term (a) is related to $\Phi$ and we have $(a) \leq R_{\max}$

Denote $\Psi_A(l) = \sqrt{\frac{(1+l)^2 l^2}{4}\|\mu_A\|_2^2 + \text{tr}(\sigma_A^2 \frac{(1+l)l}{2}I)}$ and $\Psi_H(l) = \sqrt{\frac{(1+l)^2 l^2}{4}\|\mu_H\|_2^2 + \text{tr}(\sigma_H^2 \frac{(1+l)l}{2}I)}$. For ease of presentation, we use notation $\Psi_v = \Gamma\mu_{v,0} + \Lambda(s_{\max}M\sigma_A^2 + (s_{\max} + \lambda)\|\mu_A\|^2)$ to represent the

regret term in Theorem 3 and $\Xi_H = s_{\max} M \cdot \sigma_H^2 + (s_{\max} + \lambda_H)\|\mu_H\|^2$ to represent the term in Theorem 4. Hence, building upon our results in Theorem 3 and Theorem 4, we give the upper bound of $\mathcal{R}_{A-H}(T)$

$$\mathcal{R}_{A-H}(T) \leq \sum_{l=1}^{L}(V_{\max} + lR_{\max})\gamma^l(2\Psi_A(l) + \Psi_H(l)) + \Xi_H + \frac{1}{T}\gamma^L\Psi_v$$

## H    EXPERIMENTAL SETTINGS.

In this section, we include the detailed parameter setup when conducting the experiments. The default setting is as follows:

- $\gamma = 0.85$
- $L = 5$
- $\mu_{v,0} = 10$
- $V_{\max} = 10$
- $R_{\max} = 1$
- $\mu_A = 1.8$
- $\sigma_A = 1$
- $M = 10$

In Figure 4 we choose the parameters as follows:

- $\gamma = 0.5$
- $T = 5$
- $V_{\max} = 10$
- $R_{\max} = 1$
- $\sigma_A = 0.1$
- $\sigma_H = 0.1$
- $s_{\max} = 2$
- $\lambda = 10$
- $M = 10$
- $l = 2$

We list the parameter settings of Figure 1a, Figure 1b and Figure 2a in Table 1, Table 2 and Table 3, respectively.

| Parameter | Setting 1 | Setting 2 | Setting 3 | Setting 4 | Setting 5 |
|---|---|---|---|---|---|
| $\gamma$ | 0.85 | 0.85 | 0.85 | 0.85 | 0.55 |
| $\mu_{v,0}$ | 10 | 10 | 10 | 10 | 10 |
| $V_{\max}$ | 10 | 10 | 20 | 10 | 10 |
| $R_{\max}$ | 1 | 5 | 1 | 1 | 1 |
| $\mu_A$ | 0.8 | 1.8 | 1.8 | 1.8 | 1.8 |
| $\sigma_A$ | 1 | 1 | 1 | 1 | 1 |
| $M$ | 10 | 10 | 10 | 10 | 10 |

Table 1: Parameter Settings in Figure 1a

| Parameter | Setting 1 | Setting 2 | Setting 3 | Setting 4 | Setting 5 |
|---|---|---|---|---|---|
| $\gamma$ | 0.85 | 0.85 | 0.85 | 0.85 | 0.55 |
| $\mu_{v,0}$ | 10 | 10 | 10 | 10 | 10 |
| $V_{\max}$ | 10 | 10 | 20 | 10 | 10 |
| $R_{\max}$ | 1 | 5 | 1 | 1 | 1 |
| $\mu_A$ | 0.8 | 1.8 | 1.8 | 1.8 | 1.8 |
| $\sigma_A$ | 1 | 1 | 1 | 1 | 1 |
| $M$ | 10 | 10 | 10 | 10 | 10 |
| $T$ | 10 | 5 | 5 | 5 | 5 |

Table 2: Parameter Settings in Figure 1b

| Parameter | Setting 1 | Setting 2 | Setting 3 | Setting 4 | Setting 5 |
|---|---|---|---|---|---|
| $\gamma$ | 0.85 | 0.85 | 0.85 | 0.85 | 0.55 |
| $\mu_{v,0}$ | 10 | 10 | 10 | 10 | 10 |
| $V_{\max}$ | 10 | 10 | 20 | 10 | 10 |
| $R_{\max}$ | 1 | 5 | 1 | 1 | 1 |
| $\mu_A$ | 0.8 | 1.8 | 1.8 | 1.8 | 1.8 |
| $\sigma_H$ | 0.1 | 0.5 | 0.1 | 0.1 | 0.1 |
| $\sigma_A$ | 1 | 1 | 1 | 1 | 1 |
| $M$ | 10 | 10 | 10 | 10 | 10 |
| $T$ | 5 | 10 | 10 | 10 | 10 |

Table 3: Parameter Settings in Figure 2a

