# OpenReview forum: "Heterogeneous Decision Making towards Mixed Autonomy: When Uncertainty-aware Planning Meets Bounded Rationality"
_ICLR.cc/2024/Conference — Submitted to ICLR 2024_

### Official Review · Reviewer_Sgxf · 2023-10-30

**Soundness:** 1 poor
**Presentation:** 2 fair
**Contribution:** 2 fair
**Rating:** 3
**Confidence:** 2

**Summary:**

The paper considers heterogeneous decision-making by human-driven and automated vehicles in 2 player game scenarios. The paper introduces two main lines of inquiry. The first hypothesis (H1) is understanding the relationship between learning performance and HV’s bounded rationality and AV’s planning horizon. The second hypothesis (H2) relates the impact of decision–making strategies on the overall learning performance. To evaluate H1, the paper provides a theoretical analysis through a derivation of the upper bound of the regret. To evaluate H2, the authors provide an empirical ablation over hyperparameters.

**Strengths:**

+ The paper provides a formalization of a problem that has not had extensive theoretical analysis.
+ The introduction shows this has a useful application despite being a smaller setup (2-player).
+ There is a clear description of the method assumptions and limitations throughout the paper.

**Weaknesses:**

-The main argument of the paper assumes that identifying and formalizing this problem is a significant contribution.

-Since there are only two agents, there may be methods outside of autonomous driving that may be relevant. [1] may be a good starting point to find such literature.
[1] Natarajan, M., Seraj, E., Altundas, B., Paleja, R., Ye, S., Chen, L., ... & Gombolay, M. (2023). Human-Robot Teaming: Grand Challenges. Current Robotics Reports, 1-20.

-There are many bounds proved, but it is unclear what the intuition is. Is this bound good? Could it be better? The paper is quite difficult to understand the takeaways from these bounds. This weakness would no longer be an issue after clarifying the writing.

**Questions:**

How should these theoretical results be interpreted? It would be beneficial to the reader to add some intuition regarding whether this bound is tight enough for performance.

What is the main takeaway from the empirical performance? There exists a set of parameters for a setting to minimize the regret, but then how should a practitioner use these results? Can they be extended beyond a 2-player setting or do these hold empirically with humans? Many of these results ask more questions (due to the nature of a paper formalizing a problem). Thus, I would expect much of the paper to discuss the implications of these results, which it is currently lacking.

Some minor points:
The metaphor in line 1 was a bit confusing.

Many of the abbreviations in the introduction should be introduced before using the abbreviated form. What is “NHTSA”, “HV” (only defined in abstract, may do well to just repeat once in the introduction such as Automated Vehicle (AV).

In Section 5, line 2, I believe “Av” should be “AV”

Please link to the proofs in the appendix from the main document.

---

> ### Author Response · Authors · 2023-11-15
> **Response to Reviewer Sgxf (1/2)**
>
> We thank reviewer Sgxf for your thorough review and useful feedback. We address your concerns as follows.
>
> **(Weakness)**
>
> **W1. Significance - Why our problem formulation is different from others?**
>
> **A1.** In this work, we aim to understand the complicated interactions between human and machine decision-making and develop a *fundamental understanding* of the impact of *different decision making strategies* on the system efficiency (in terms of dynamic regret). To this end, our problem formulation considers  different learning approaches  for each agent's decision making. Specifically, we compare the major differences with other relevant problem formulations, e.g., Dec-POMDP, as follows.
>
>
> - Many MARL formulations for  Dec-POMDP, often assume all agents   use the *same* RL algorithms to 'maximize' the rewards. Our work aims to study the case where AV and HV  use different learning methods, i.e., longer-horizon look-ahead planning and myopic decision making, to achieve their objectives.
> - Further,  in our setting, HV makes decision  with bounded rationality at each time step, which *deviates from reward maximization*.
> - (Theoretical Results) In our setting, we study the regret dynamics of the system (cf. $\mathcal{R}\_{A-H}$  Section 4) such that the impact of different learning strategies on the system performance is characterized. Specifically, we show how HV's bounded rationality, AV's planning horizon and function approximation error have impact on the overall system dynamics. We remark this is different from the analysis in MARL formulation, where often times the objective is to characterize the Nash Equilibrium. Thus, the analysis method used in our work is very different from previous MARL formulations.
>
>
> **W2. Relevant works on two agents interaction**
>
> **A2.** We thank the reviewer for pointing out the very interesting paper on Human-Robot Teaming [R1]. In what follows, we compare our work with [R1] and the reference therein  and outline the new ideas beyond the literature.
>
>
> - (Bounded Rationality of HV's Decision Making.) In the same spirit as in [85-86, 104-108], we consider the human's decision making to be sub-optimal (cf. Section 3.2). More importantly, we study the impact of different bounded rationality levels on the overall system performance (cf. Figure 3(c)).
> - (Theoretical Study on HV-AV Interaction.) Despite the rich empirical results [89,154],   the theoretical analysis on the interaction between AV and HV is still lacking, especially considering their different decision making.   Our work aim to obtain a fundamental understanding of the heterogeneous decision making in the interplay, especially the impact of HVs' decision making with bounded rationality on AVs' performance, which is crucial for achieving efficient mixed autonomy.
> - (Problem Formulation) Our problem formulation considers AV's modeling of HV's behaviors [86-90] (ref.~Equation (2)). Meanwhile, in our work, we use regret dynamics as a metric to "evaluate the human experience" [298-301], i.e., we show how does HV's learning strategies have impact on the overall learning performance by characterizing the system regret $\mathcal{R}\_{A-H}$ (ref. Section 4).
>
> [R1] Natarajan et al., "Human-Robot Teaming: Grand Challenges". Current Robotics Reports, 1-20. (2023)
>
> **W3. What is the intuition of the bounds? Is this bound good? Could it be better?**
>
> **A3.** To improve the presentation, we reorganize the description and analysis of our theoretical results in our revision; and the changes are marked in blue. We summarize the main changes as follows. We provide the intuition of these bounds in our answer to Question 1 below.
>
> - (Lemma 1 and Lemma 2.) We clarify that the derived performance gaps are tight.
>
> - (Theorem 3.) We provide a more detailed explanation of each parameter in the upper bound of the regret and clarify the bound is tight.
>
> - (Theorem 4.) We add the analysis of the upper bound for HV's regret and clarify the bound is tight.
> - (Corollary 5) We provide a more detailed discussion on the coupling between AV's decision making and HV's decision making.

---

> ### Author Response · Authors · 2023-11-15
> **Response to Reviewer Sgxf (2/2)**
>
> **(Questions)**
>
> **Q1. How to interpret those theoretical results?**
>
> **A1.** We clarify "the intuition regarding whether this bound is tight enough for performance" as follows.
>
> - (Lemma 1 and Lemma 2.) Both bounds on the performance gap are tight. It can be seen that the upper bound equal to zero when we set $\mu\_A = \sigma_A=0$ (such that AV's prediction on HV's action is accurate), the upper bound is only related to the function approximation error term $\gamma^L\mu\_{v,t}$. Furthermore, by setting $L \to \infty $ (such that the Q-function does not rely on the terminal cost approximation), the upper bound equal to zero which indicating there is no performance gap.
> - (Theorem 3.) We clarify the upper bound for AV's regret is tight, i.e., by setting $\mu_A = \sigma\_A=0$, the residual term $(\gamma^L/T) \Gamma \mu\_{v,0}$ is only related to the function approximation error, which will be close to zero as we increase $L$.
> - (Theorem 4.) We clarify the bound is tight since the upper bound will be zero once we set $\mu\_H = \sigma\_H=0$. Intuitively, HV's regret is solely due to its bounded rationality.
> - (Corollary 5) The upper bound is tight. We first set $\mu\_A = \sigma\_A=\Psi\_v = 0$, then the residual terms are only relevant to  HV's bounded rationality $\Psi\_H,\Xi\_H$. Now let $\mu\_H = \sigma\_H=0$, we obtain the zero regret of the whole system.
>
>
>
> **Q2. What is the main takeaway from the empirical performance? How should a practitioner use these results?**
>
> **A2.** In what follows, we summarize the main takeaway, as supported by the empirical experiments illustrated in Figures 1-4.
>
> - (**How to adjust look-ahead length** $L$?). In Figure 1 and Figure 2(a), we show the Goodhart's Law in AV's learning performance and it indicates that  increasing the planning horizon will initially help   the learning performance until a critical point.  In practice, we remark that adjusting the look-ahead length (e.g., through grid search) is essential to help AV achieve the desired performance.
> - (**How to choose discounting Factor** for AV?) In Figure 2(b), we show that the discounting factor should be set in a way that considers the function approximation error. For instance, if the function approximation error is more dominant than the prediction error, the small discounting factor tends to help to reduce the regret.
> - (**Priority of training**: function approximation v.s. prediction model?) In Figure 3(a-b), we show the impact of different function approximation error ($\mu\_{v,0}$) and prediction error ($\mu\_A$) on the system regret. It can be seen that by reducing the prediction error from 0.4 to 0.2, the regret have significant change from 4000 to 1800 (-$30\%$). While reducing the function approximation error from 100 to 50, the regret changes from 51.41 to 51.38 (-$0.06\%$). The empirical results indicate that optimizing over prediction model tends to help us get more improvement on regret.
>
>
> **Q3. Extend to more than two-player settings? Or do these hold empirically with humans?**
>
> **A3.** We remark that it is feasible to extend to more than one AV and one HV setting and  we share some preliminary thoughts as follows. Assume there are $N\_H$ HVs and $N\_A$ AVs in the mixed traffic system. With abuse of notations, we define the action vector for AVs and HVs as follows, at time step $t$,
> $$u_H(t) = [u_{H,1}(t), u_{H,2}(t),\cdots, u_{H,N_H}(t)]$$
>
> $$u_A(t) = [u_{A,1}(t), u_{A,2}(t),\cdots, u_{A,N_A}(t)]$$
>
>
> By defining the prediction error as in Equation (4) and HVs' bounded rationality as in Section 3.2, our analysis framework still can be applied. We remark that the dimenion of the approximation error term and the bounded rationality term is thus $N\_A$ and $N\_H$ times higher than the two-agent case. Hence, the resulting regret in Theorem 3 and Theorem 4 are $N\_A$ and $N\_H$ times higher than the two-agent case.
>
>
> **Q4. Minor points: metaphor in line 1 was confusing? Abbreviations issue. Link to the appendix.**
>
> **A4.** We thank the reviewer to point out those minor issues and our revision has changed in the following ways,
>
> - We add reference for the metaphor in line 1.
> - We correct the abbreviation issues in the paper.
> - We add the link to the appendix in the main paper.

---

> ### Author Response · Authors · 2023-11-18
> **Thank Reviewer Sgxf for your review**
>
> Thank you Reviewer Sgxf again for your detailed review.
>
> Since the final stage of the discussion between reviewers and authors will end soon, please let us know if you have any further comments on our response to your concerns, we will be more than happy to answer your questions.

---

### Official Review · Reviewer_DRuU · 2023-11-01

**Soundness:** 3 good
**Presentation:** 4 excellent
**Contribution:** 2 fair
**Rating:** 6
**Confidence:** 2

**Summary:**

The paper delves into the challenges and advancements associated with mixed autonomy in traffic scenarios, where both automated vehicles (AVs) and human-driven vehicles (HVs) coexist.
The primary objective of the study is to gain insights into the interactions between HVs and AVs by framing these interactions as a process wherein AVs predict the decisions made by HVs.
Regret analysis serves as a metric for evaluating the learning performance of both systems, and the authors conduct a comprehensive analysis to unveil how the overall system performance is influenced by HVs' bounded rationality and AVs' planning horizon.

**Strengths:**

The reviewer has limited familiarity with the field of autonomous driving research. Based on the reviewer's best judgment:

- The paper offers a novel perspective on mixed autonomy, emphasizing the diverse decision-making processes involving both human and machine behaviors.

- The paper's introduction and problem formulation are well-defined and effectively presented. The paper's structure is logical and facilitates easy comprehension.

- The formulation of interactions between human-driven vehicles (HVs) and automated vehicles (AVs) appears innovative and of significant relevance. The regret analysis applied to both systems is robust, and it is intriguing to observe how the results reveal the overall system performance's dependence on HVs' bounded rationality and AVs' planning horizon.

**Weaknesses:**

The study relies on a set of assumptions such as:

- Both HVs and AVs operate within the same action space;

- HVs plan actions over a shorter time horizon, whereas AVs adopt a longer horizon for decision-making;

The practicality of these assumptions raises questions.

The formulation of HV-AV interactions leans on Goodhart’s law to evaluate AV performance and incorporates a model wherein HVs react to AVs based on bounded rationality. At this stage, I am unable to comment on the robustness of such a formulation.

Some comparative experiments and analysis would strengthen the paper's contributions.

I will refine my viewpoint upon reviewing the feedback from other reviewers.


Minor:

Please use the ICLR2024 template.

**Questions:**

Could the authors elaborate on the foundational assumptions made for the formulation presented in the paper？

---

> ### Author Response · Authors · 2023-11-15
> **Response to Reviewer DRuU (1/2)**
>
> We thank reviewer DRuU for your thorough review and positive comments. We address your concerns as follows.
>
> **(Weaknesses)**
>
> **W1. The practicality of the assumptions.**
>
> **A1.** We clarify a few points in  our study as follows.
>
> - (Action Space for HV and AV.) We clarify that we do not assume both vehicles have the same action space. As can be seen in Section 2, we denote the action space for AV and HV to be $\mathcal{U}\_A$ and  $\mathcal{U}\_H$, respectively. We do assume that both vehicles' action space have the same cardinality. Intuitively, though different vehicles can drive forward, backward, parking (the cardinality of the action space are both 3), the corresponding actions (Action space) can be different, e.g., the wheel angle, forces on the brake, etc.
> - (What is the intuition of "HV plans actions over a shorter time horizon, whereas AVs adopt a longer horizon for decision-making"? ) We clarify that the difference between human decision making and  AVs decision making stems from the inherent differences such as cognition systems and learning/computing capabilities. Our problem formulation of human decision making is related to the studies in psychology and  behavior theory [R1,R2,R3], where the main discovery on human's decision making process is that "human individuals have constraints in both their understanding of their surroundings and their computational capacities" (ref. Paragraph 2 in Introduction and Section 2.1). To this end, we model human as a "myopic" decision maker by only plan over $N$-step time horizon. We remark that we do not have constrains on the maximum value of $N$ in our analysis. Meanwhile, the AVs are generally equipped with sensing, storage and computing system on board such that AVs are able to handle complex optimization problem in the real-time, e.g., combing the information from camera, radar to make prediction of the other vehicles' future actions. The advantage of the computation capability can also help with long-term planning. For example, by taking advantage of the traffic information (e.g., interactive maps with congestion prediction, road closure, etc.), AVs are able to plan a longer trajectory than HVs, who generally are making the decision with limited local information [R4].
>
> [R1] Herbert A Simon. Models of man; social and rational. 1957.
>
> [R2] Herbert A Simon. Rational decision making in business organizations. The American economic review, 69(4):493–513, 1979.
>
> [R3] Daniel Kahneman. Maps of bounded rationality: Psychology for behavioral economics. American economic review, 93(5):1449–1475, 2003.
>
> [R4] Song et al. "Pip: Planning-informed trajectory prediction for autonomous driving." ECCV, 2020.

---

> ### Author Response · Authors · 2023-11-15
> **Response to Reviewer DRuU(2/2)**
>
> **W2. Robustness of the problem formulation?**
>
> **A2.** We thank the reviewer's interest in the robustness of our problem formulation. We clarify that our problem formulation is general and considers the practical interaction between HVs and AVs.
>
> - (**Adaptive Interaction between HVs and AVs.**) In this work,  we consider both AV and HV adapt their policies during the interaction. More specifically, as shown in Section 2.1 Equation (2) and Equation (3) (and the paragraph below), both AV and HV's policy is conditioned on each other and will update its policy when the other agent has changed its policy. We remark that this formulation corroborates with the real-world human-autonomous vehicle interaction.
> - (**General Formulation for AV and HV's Decision Making** (cf. Equation (2), Equation (3)).) In Equation (2), we consider AV's decision making can be either model free (by setting $L=1$) or model-based (by setting $L>1$). Meanwhile, we do not constrain the method on deriving the policy $\pi$ such that HV's learning method can be either $Q-$learning (greedy policy) or Actor-Critic (by using policy gradient). Meanwhile, in Equation (3), we consider AV's decision making to be $N$-step planning while we do not impose any constrains on the length of $N$. In particular, when $N\to \infty$, the decision making of HV is related to dynamic programming (assume the model is available) and otherwise, the decision making of AV is in the same spirit of Model Predictive Control (MPC).
> - (**Beyond Two-agent Case**) We remark that it is feasible to extend to more than one AV and one HV setting and  we share some preliminary thoughts as follows. Assume there are $N\_H$ HVs and $N\_A$ AVs in the mixed traffic system. With abuse of notations, we define the action vector for AVs and HVs as follows, at time step $t$, $u\_H(t) = [u\_{H,1}(t), u\_{H,2}(t),\cdots, u\_{H,N_H}(t)]$ and $u\_A(t) = [u\_{A,1}(t), u\_{A,2}(t),\cdots, u_{A,N\_A}(t)]$. By defining the prediction error as in Equation (4) and HVs' bounded rationality as in Section 3.2, our analysis framework still can be applied. We remark that the dimension of the approximation error term and the bounded rationality term is thus $N\_A$ and $N\_H$ times higher than the two-agent case. Hence, the resulting regret in Theorem 3 and Theorem 4 are $N\_A$ and $N\_H$ times higher than the two-agent case.
>
>
> **W3. Comparative experiments and analysis**
>
> **A3.** We thank the reviewer for the suggestions on strengthen this paper's contribution. In our revision, we make the following changes in this regard:
>
> - (Comparison with conventional multi-agent problem formulation.) We further clarify the difference between our problem formulation and other multi-agent formulation (e.g., MARL, Dec-POMDP, Two-player game) and emphasize the robustness of our formulation in Section 1 (related work) and Section 2.
> - (Detailed analysis on the theoretical results.) We  add  insight on the theoretical results and focus on the impact of each key elements of the decision makings (e.g., look-ahead length $L$, prediction error, etc.) in Section 3 and Section 4.
> - (Thorough analysis on the empirical results.) We emphasize the comparison among different parameter setting in Figures 1-3 and clarify the main takeaway   for  algorithm design from a practitioner perspective, in Section 4.

---

> ### Author Response · Authors · 2023-11-18
> **Thank Reviewer DRuU for your review**
>
> Thank you Reviewer DRuU again for your detailed review.
>
> Since the final stage of the discussion between reviewers and authors will end soon, please let us know if you have any further comments on our response to your concerns, we will be more than happy to answer your questions.

---

> > ### Comment · Reviewer_Sgxf · 2023-11-22
> >
> > I thank the authors for the detailed responses. While I disagree regarding the 2-agent vs 3+ agent significance, I make my decision based on the clarity of the 2-agent case. Despite substantial effort on the author's behalf, I find the overall paper writing still very confusing. I agree with reviewer dAQk's claims about clarity and will keep my score as is.

---

> > > ### Author Response · Authors · 2023-11-22
> > > **Reply to Reviewer Sgxf's Response**
> > >
> > > We thank the reviewer for your feedback. Would you please clarify your disagreement "regarding the 2-agent vs 3+ agent significance"?

---

### Official Review · Reviewer_dAQk · 2023-11-01

**Soundness:** 1 poor
**Presentation:** 2 fair
**Contribution:** 1 poor
**Rating:** 3
**Confidence:** 3

**Summary:**

This paper is concerned with a setting where autonomous vehicles must deal with humans that have bounded rationality. Under this scenario, this work provides a theoretical upper bound on the resulting regret of a planning-based autonomous vehicle with a finite planning horizon. The upper bound is then utilized to analyze the impact of various factors (e.g. the human's bounded rationality, the autonomous vehicle's finite planning horizon, the discount rate used during planning, etc.) towards the regret upper bound produced by the human-autonomous vehicle interaction. Finally, the authors claimed that they use their proposed upper bound formulation to analyze the impact of different learning strategies in human-autonomous vehicle interaction.

**Strengths:**

**Major Strength - Originality - Novelty of Regret Analysis**

To the best of my limited knowledge of the subject, I have not seen a regret analysis in a human-autonomous vehicle interaction where humans have bounded rationality. While the proposed analysis seems to be based on very limiting assumptions, it could provide an important basis for future works in human-autonomous vehicle interaction. Perhaps from a multiagent systems researcher's perspective, the closest work to this paper is from Loftin & Oliehoek (2022), who also explored regret bounds when dealing with adaptive partners (that can be humans). Nonetheless, this work does not account for a human partner's bounded rationality in their theoretical analysis and is limited to fully cooperative settings.

**Major Strength - Clarity**

In general, I find the paper to be clearly written. I especially appreciate the authors pointing out which factors of the human or autonomous vehicle's decision-making each theorem/analysis addressed. That helped the readers to stick with the overall flow of the paper, which is always challenging with papers proposing lots of theoretical analysis.

Citations:

Loftin & Oliehoek. 2022. "On the Impossibility of Learning to Cooperate with Adaptive Partner Strategies in Repeated Games". ICML 2022.

**Weaknesses:**

**Minor Weakness - Quality - Missing Citations to Related Works**

While not limited to their applications to autonomous vehicles, there are works in multiagent systems concerned with interaction against partners (including humans) whose decision-making processes (which encompass many things, such as their beliefs, goals, policies, bounded rationality, etc.) are unknown. These works can usually be found in the literature on ad hoc teamwork [1], zero-shot coordination [2], or opponent modelling [3].  Following how human-autonomous vehicle interaction can be modelled using the formulation described in these research areas, there should be a reference to these works in the paper.

**Minor Weakness - Quality - Problem Formulation**

The decision to model the interaction as an MDP is also questionable. It specifically goes against the original formulation of MDPs to have the environment output two reward functions (one function each for the human and autonomous vehicle). In this case, would it not have been better to model the problem as a Dec-POMDP (or any other multiagent systems formulation)?

**Major Weakness - Quality - Non-Adaptive Policy Assumption**

As highlighted by the authors themselves, a major weakness of this paper lies in the assumption that human and autonomous vehicle policies do not change/adapt during interaction. This assumption plays an important role in the analysis where teammate actions are assumed to be fixed when establishing the regret upper bounds of the autonomous vehicle's decision-making process. But this, overall, seems like an unrealistic assumption in real-world human-autonomous vehicle interaction.

In general, I highly recommend the authors to consider the concept of adaptive regret [4] in their analysis. Unlike the notion of regret used in this work, adaptive regret accounts for the possible change in a partner's decision-making process during interaction.

**Minor Weakness - Quality - Misrepresented Contribution**

In the final sentence in the introduction, the authors claimed that they investigated the effects of different learning strategies on the learning performance achieved through the human-autonomous vehicle interaction. Unfortunately, the analysis in Section 4 did not exactly achieve that and instead settled with analyzing the effects of an autonomous vehicle's approximation errors on the overall regret. If the authors wanted to stick with the original claim, it would have been better if they formulated different types of learning algorithms and analyzed the effects on the regret bounds (see Section 5 of [4]). Otherwise, the authors should adequately adjust the reader's expectations in the introduction by being more direct about which factors' effects on the regret bounds are analyzed.

**Major Weakness - Significance**

I am concerned with the significance of this work. First, it seems based on highly restrictive assumptions that would not hold in real-world scenarios, where humans will adapt to the autonomous vehicle's policy. While this could have been fine had there been no previous theoretical analysis on dealing with adaptive partners, Loftin and Oliehoek (2022) [4] have introduced concepts for this kind of analysis. To improve the significance of this work for the community, I would expect the authors to build their analysis from the adaptive regret concept [4].

At the same time, there isn't a sufficient analysis of the effects of different autonomous vehicle learning algorithms on the overall regret in this paper. This generally limits the usefulness of this work for other people working on learning algorithms for autonomous vehicles. All in all, I doubt this work can significantly influence future work in this field.

Citations:

[1] Mirsky et al. (2022). "A Survey of Ad Hoc Teamwork Research". EUMAS 2022

[2] Hu et al. (2020). "Other-Play for Zero-Shot Coordination". ICML 2021.

[3] Albrecht et al. (2017). "Autonomous Agents Modelling Other Agents: A Comprehensive Survey and Open Problems". AIJ

[4] Loftin & Oliehoek. 2022. "On the Impossibility of Learning to Cooperate with Adaptive Partner Strategies in Repeated Games". ICML 2022.

**Questions:**

1. What are the challenges to extending the existing theoretical analysis to an adaptive setting where both human and autonomous vehicle change their policies during an interaction?

2. Can this analysis be extended to other learning strategies that are not model-based?

3. Why is it appropriate to model the system as an MDP? Doesn't the environment output two rewards at each timestep (which goes against the definition of MDPs)?

4. Can you explain why the non-linear model (Section 3) is suitable to model human-autonomous vehicle interaction?

**Details Of Ethics Concerns:**

I do not have ethical concerns over the paper

---

> ### Author Response · Authors · 2023-11-15
> **Response to Reviewer dAQk (1/4)**
>
> We thank reviewer dAQk for your thorough reading and constructive criticism. We answer your questions and address your concerns as follows.
>
> **(Weakness)**
>
> **W1. Minor Weakness-Missing Citations**
>
> **A1.** We thank the reviewer for pointing out references [1-3], and we have added a comparison between these works and ours in Section 1 (Related Work), which is outlined below.
>
>
> - Ad-hoc teamwork problem [1] mainly focused on the cooperative case,  whereas our setting does not impose assumptions on the cooperation of agents.
>
> - Zero-shot coordination [2] focused on the robustness of the self-play, whereas our work aims to understand the interaction between two agents with different decision makings strategies.
>
> - Opponent modeling [3] addressed the methods on modeling the opponents, whereas our work focuses on characterizing the impact of such modeling errors on the learning performance.
>
>
> **W2. Minor Weakness-Problem formulation goes against MDP? Why not Dec-POMDP?**
>
> **A2.** In fact,  our  formulation can be viewed as a  MDP in the following sense: From the whole system's perspective, let $u=[u\_A,u\_H]$ to be the action vector which consists the action of both AV and HV. Let $r=[r\_A,r\_H]$ be the reward vector that consists the reward of both vehicles. In this case, we have the system MDP as $\langle \mathcal{S},\mathcal{U}, P, r, \gamma \rangle$.  In a nutshell, our MDP formulation considers a multi-agent RL system with independent learners [R1].  We  outline a few new ideas beyond the conventional Dec-POMDP  as follows.
>
> - Many MARL formulations for  Dec-POMDP, often assume all agents   use the same RL algorithms to 'maximize' the rewards. Our work aims to study the case where AV and HV  use different learning methods, i.e., longer-horizon look-ahead planning and myopic decision making, to achieve their objectives.
>
> - Furthermore,  in our setting, HV makes decision  with bounded rationality at each time step, which deviates from reward maximization.
>
> [R1] Zhang et al., "Multi-agent reinforcement learning: A selective overview of theories and algorithms." Handbook of reinforcement learning and control (2021): 321-384.

---

> ### Author Response · Authors · 2023-11-15
> **Response to Reviewer dAQk (2/4)**
>
> **W3. Major Weakness-AV and HV's policy is not adaptive?**
>
> **A3.** We thank the reviewer for sharing the very interesting paper on adaptive regret  [4].  We clarify that in fact, both AV and HV  **adapt their policies** during the interaction. More specifically, as shown in Section 2.1 Equation (2) and Equation (3) (and the paragraph below), both AV and HV's policies are conditioned on each other and each agent will update its policy when the other agent has changed  its policy, summarized below:
>
>
> - (HV adapts its policy via $N$-step planning.) In the same spirit as in Model Predictive Control (MPC), HV chooses the action by planning ahead for $N$ steps (cf. Equation (3)), i.e., at each time step $t$, HV's (optimal) policy is given by $$\pi\_{H}(\cdot \vert x(t),u\_A(t)): ~u\_H^{\*}(t):=\arg\max\_{u\_{H}(t)}\max\_{u\_{H}(t+1),\cdots, u\_{H}(t+N-1)} \Phi\_H(x(t),u\_A(t),u\_H(t)),$$
>   where the policy is conditioned on (predicted) AV's action $u\_A(t)$ from time $t$ to $t+N-1$. Along the line of (Sadigh et al. (2016,2018)), we remark that predicting a short-term sequence of controls is manageable for human, i.e., $u\_A(t)$ is the true action taken by AV. At the next time step $t+1$, HV's policy is further adapted to AV's policy through its prediction of AV's future action $u\_A(t+1),\cdots,u\_A(t+N)$.
>
> - (AV adapts its policy via $L$ -step lookahead RL.) Similarly, AV's policy is derived by maximizing the $L$ -step lookahead planning objective (ref. Equation (2)), i.e., at each time step $t$,
>   $$\hat{\pi}\_A^t = \hat{\pi}\_A(\cdot \vert x(t),u\_H(t)) = \arg \max\_{u\_A(t)} \max\_{u\_A(t+1),\cdots,u\_A(t+L-1) } \hat{Q}\_{t}(x(t),u\_A(t),\hat{u}\_H(t)).$$
>   We note that AV's policy is conditioned on (predicted) HV's action $\hat{u}\_H(t)$, which is a noisy prediction of the underlying true action ${u}\_H(t)$ (ref. Equation (4)). In this way, AV's policy will adapt to HV's policy change.
>
> **(Relation to Adaptive Regret)** Moreover, we clarify that in fact our regret analysis  uses **adaptive regret** which "accounts for the possible change in a partner's decision-making process during interaction". For instance, in our regret definition for HV (ref. Section 3.2), let ${\pi}\_A^t = {u}\_A(t)$ and $\pi\_H^t=u\_H(t)$ be the policies followed by AV and HV at time $t$. By abuse of notations,
> $$\Phi\_H^{\*}(t) := \Phi\_H(x(t),\pi\_H^{\*}(t),\pi\_A(t)) = \max\_{\pi\_H \in \Pi\_H}\Phi\_H(x(t),\pi\_H,\pi\_A(t)).$$
>
> Then we obtain the regret of HV as
> $$\mathcal{R}\_H(T\vert \pi\_A) = \frac{1}{T} \mathbf{E}\_{x}[\sum\_{t=1}^T (\max\_{\pi\_H \in \Pi\_H} \Phi(x,\pi\_A^t,\pi\_H)-\Phi(x,{\pi}\_A^t, \pi\_H^t))].$$
> Notice that at each time step $t$, AV does not follow a fixed policy. Meanwhile, HV's optimal policy is determined by choosing a policy $\pi\_H^{\*}(t)$ from policy space $\Pi\_H$ such that $\Phi(x,\pi\_A^t,\pi\_H)$ is  maximized. We remark that the definition of the regret given in our work is related to the one given in Equation (2)[4]. We clarify the relation between the two definitions in our revision.
>
> **(Regret Dynamics)** Moreover, in **Section 4**, we present the analysis of the regret dynamics, i.e., how does the regret change over interaction time $t$ (ref. Fig. 4). It can be seen from Fig. 4 that the regret dynamics of the overall system, i.e., $\mathcal{R}\_{A-H}$, is changing during the interaction. Note that this is different from the the external regret (e.g., Equation (3) [4]), where $G(a\_n,b\_n),n=1,2,\cdots,s$ is the one-step payoff. In our case,  $Q$ and $\Phi$ are the multi-step look-ahead or planning objective function when HV follows policy $\pi_H^t$ and AV follows policy $\pi\_A^t$, $t=1,2,\cdots,T$.

---

> ### Author Response · Authors · 2023-11-15
> **Response to Reviewer dAQk(3/4)**
>
> **W4. Minor Weakness-Misrepresented Contribution?**
>
> **A4.** We clarify that by saying "different learning strategies", we refer to the setting where *AV's learning strategy* (or "learning algorithm") is different from *HV's* during the interaction, i.e.,
>
> - AV uses $L$-step look-ahead planning (cf. Eqn. (2))
> - HV uses $N$-step planning (cf. Eqn. (3)).
>
> In Section 4, we show the impact of AV and HV's learning strategies have impact on the overall system performance in terms of the regret of the whole system $\mathcal{R}\_{A-H}$. In particular,
>
>
> - **Impact of AV's $L$-step look-ahead planning** (related to $\Psi\_A,\Psi\_v$ and look-ahead length $L$ in Corollary 5). Notably, $\Psi\_A$ and $\Psi\_v$ represent not only the impact of AV's function approximation error (ref. Figure 3(a)) but also AV's prediction error on HV's action (ref. Figure 3(b)). Moreover, $L$ plays a critical role in AV's planning algorithm and at time $t$, the look-ahead length is not directly relevant to AV's function approximation error $\hat{Q}\_{t-1}$ (it will have impact on the function approximation error at next time step).
> - **Impact of HV's $N$-step planning** (related to $\Psi\_H, \Xi\_H$ in Corollary 5).  HV's decision making hinges heavily on its bounded rationality level (i.e., $\mu\_H$). To this end, the $\Psi\_H, \Xi\_H$ terms in the upper bound characterize the impact of the rationality level on the overall system performance. Further more we demonstrate the impact of HV's rationality level in Figure 3(c).
> - **AV-HV's Impact on Each Other**.  The coupling term $\gamma^l \Psi\_H(l)$  shows that AV's decision making (e.g., $L$) can directly affect HV's impact on the system performance, e.g., either weaken or intensify HV's impact on system performance.
>
> **W5. Major Weakness-Significance: Assume HV and AV have fixed policy? Consider different autonomous vehicles learning algorithms?**
>
> **A5.** As clarified in A3 above,  our HV-AV interaction setting (cf. Equation (2), Equation (3)) indeed considers the case where both vehicles adapt to the other's policy.
>
> Next, we remark on the rationale of choosing $L$-step look-ahead planning as AV's learning method in our analysis, i.e.,
>
> - Why model-based planning? In the context of AV-HV interaction, the system model represents the state (e.g., locations of two vehicles) transition, which follows the physics laws and can be learned by using offline dataset. Comparing with model-free learning, model-based method is more data efficient and thus more practical for real-world applications [R1]. Moreover, one of the advantage of model based RL is the prediction capability, i.e., predict the future state by rolling out the model. The prediction capability of autonomous vehicles has been proved to be useful for safety [R2].
> - Why the learning method defined in Equation (2) is in fact general?  We clarify that Equation (2) is related to many commonly used RL algorithms, for instance,
>   - (Model-free Case) Set $L=1$, Equation (2) is the model-free Q-function update and our regret analysis still holds.
>   - (Actor-Critic Case) Let $Q$-function and policy $\pi$ be parameterized by $\theta$ and $\phi$, respectively, Then Equation (2) can be learned by using Actor-Critic, i.e., in the actor step,  $\theta$ is updated by maximizing the $L$-step look-ahead objective and $\phi$ is updated using policy gradient. Note that in this case, the approximation error in both Actor and Critic update can be encapsulate into $\epsilon\_{v,t}$ as in Assumption 1.  Our proof of the regret carries over.
>
> [R1] Kiran, B. Ravi, et al. "Deep reinforcement learning for autonomous driving: A survey." IEEE Transactions on Intelligent Transportation Systems 23.6 (2021): 4909-4926.
>
> [R2] Ren, Xuanchi, et al. "Safety-aware motion prediction with unseen vehicles for autonomous driving." Proceedings of the IEEE/CVF International Conference on Computer Vision. 2021.

---

> > ### Comment · Reviewer_dAQk · 2023-11-21
> > **Response to Author Rebuttal (3/4)**
> >
> > **W4. Minor Weakness-Misrepresented Contribution?**
> >
> > I would recommend a more specific term to describe the variation of decision-making strategies compared in the analysis. Perhaps maybe changing "learning strategies" to "number of steps for lookahead planning".
> >
> > The problem here is that learning strategies, especially in the context of non-stationary teammates, can be mistaken for different ways an agent/human learns/changes their policy following the observed actions of other agents from the past. Being more specific should help avoid this misunderstanding and improve overall clarity.
> >
> > **W5. Major Weakness-Significance: Assume HV and AV have fixed policy? Consider different autonomous vehicles learning algorithms?**
> >
> > When you claim that your analysis is wrt adaptive agents, how come the model-based planning only predicts the future state of the environment? With adaptive teammates, different actions taken by the planning agent during rollout would change the policies of the other agents in different ways (after all, they see the agent playing different actions). Do the authors also model the change in teammate policies during planning (apart from just modelling the change in states)?

---

> > > ### Comment · Reviewer_dAQk · 2023-11-21
> > > **Response to Author Rebuttal (4/4)**
> > >
> > > Overall, I am keeping my scores as it is.
> > >
> > > The problem with this paper is the clarity regarding how other agents/humans' past actions affect the policy/planning process of a controlled agent. Despite the authors claiming that this work is analyzing co-adaptive agents, the proposed agent/human objective together with the model-based planning method does not clearly indicate that other agents' past actions can affect a controlled agent's objective/planning process. All things considered, there has to be significant improvement in terms of clarity, even if perhaps the results from the authors are correct.

---

> > > ### Author Response · Authors · 2023-11-22
> > > **Reply to Reviewer dAQk's response (3/4)**
> > >
> > > **W4. Minor Weakness-Misrepresented Contribution?**
> > >
> > > **A4.** Thanks for your suggestion. This is more or less a stylish issue, and we have made it very clear that by  "different learning strategies", we refer to the setting where *AV's learning strategy (or "learning algorithm") is different from HV's* during the interaction, i.e.,
> > >
> > > - AV uses $L$-step look-ahead planning (cf. Eqn. (2))
> > > - HV uses $N$-step planning (cf. Eqn. (3)).
> > >
> > >
> > > **W5. Major Weakness-Significance: Assume HV and AV have fixed policy? Consider different autonomous vehicles learning algorithms?**
> > >
> > > **A5.** We consider the Stochastic game with independent learners, where "each agent chooses best-response type actions (as opposed to trying to influence other agent as would happen in a game) to other agents' strategy without any coordination with her opponent". In particular, we use the same setting as in previous work [R1, R2] to enforce the turn-taking, in which AV predicts the robot's action prior to selecting her actions. It maintains the adaptive nature of two agent's interaction since HV and AV can influence other agent's policy, e.g., by selecting policies that force a particular response.
> > >
> > > Regarding your concerns on "how come the model-based planning only predicts the future state of the environment",  we follow the literature  on model-based planning (e.g., [R1, R2]). More specifically,
> > >
> > > - Model-based planning specifically refers to the process of using a learned or given model of the world to construct imagined **future trajectories or plans**. In our work, the planning is conditioned on the prediction of other agent's future action.
> > > - We consider the Stochastic game with independent learners, where "each agent chooses best-response type actions (as opposed to trying to influence other agent as would happen in a game) to other agents’ strategy without any coordination with her opponent". In particular, we use the same setting as in previous work [R1, R2] to enforce the turn-taking, in which AV predicts the robot's action prior to selecting her actions. It maintains the adaptive nature of two agent's interaction since HV and AV can influence other agent's policy, e.g., by selecting policies that force a particular response.
> > >
> > > [R1] Sadigh, Dorsa, et al. "Planning for cars that coordinate with people: leveraging effects on human actions for planning and active information gathering over human internal state." Autonomous Robots 42 (2018): 1405-1426.
> > >
> > > [R2] Sadigh, Dorsa, et al. "Planning for autonomous cars that leverage effects on human actions." Robotics: Science and systems. Vol. 2. 2016.

---

> ### Author Response · Authors · 2023-11-15
> **Response to Reviewer dAQk(4/4)**
>
> **(Questions)**
>
> **Q1. What is the challenge to consider the adaptive setting?**
>
> **A1.** We clarified in A3 above that our HV-AV interaction setting (cf. Equation (2), Equation (3)) indeed considers the case where both vehicles adapt to the other's policy.
> One main objective of this work is to characterize regret dynamics and understand the impact of different decision making strategies on learning performance.
>
> **Q2. Non model-based analysis?**
>
> **A2.** We clarify that the learning method defined in Equation (2) reduces to model-free Q-learning by setting $L=1$ (ref. A5 in the Weakness part above).
>
> **Q3. Why model the system as MDP?**
>
> **A3.** We clarify that our MDP formulation our MDP formulation considers a multi-agent RL model with independent learners (ref. A2 in the Weakness part above).
>
> **Q4. Why non-linear model is suitable to model the interaction?**
>
> **A4.** We note that the non-linear model is common in the real world traffic flow. For instance, the well known intelligent driver model (IDM) is initially proposed to demonstrate the general approach on smoothing nonlinear mixed traffic flow [R3].
>
> [R3] Gazis, Denos C., Robert Herman, and Richard W. Rothery. "Nonlinear follow-the-leader models of traffic flow." Operations research 9.4 (1961): 545-567.

---

> ### Author Response · Authors · 2023-11-18
> **Thank Reviewer dAQk for your review**
>
> Thank you Reviewer dAQk again for your detailed review.
>
> Since the final stage of the discussion between reviewers and authors will end soon, please let us know if you have any further comments on our response to your concerns, we will be more than happy to answer your questions.

---

> ### Comment · Reviewer_dAQk · 2023-11-21
> **Response to Author Rebuttal (1/4)**
>
> **W1. Minor Weakness-Missing Citations**
>
> Thank you for adding these new citations. I believe this would really help position the authors' work with respect to other related research problems.
>
> One minor note is that the goal of zero-shot coordination is not limited towards evaluation against agents trained via self-play. More recent works in ZSC/AHT also evaluate against agents with different decision-making strategies or even with humans. Thus, I see no difference between this work's problem setup and ZSC/AHT. Nonetheless, I would encourage the authors to clearly emphasize the theoretical aspect of their work as opposed to most ZSC/AHT works that tend to be more empirical in nature.
>
> **W2. Problem Formulation**
>
> The assertion that Dec-POMDP is limited to model agents using the same algorithm is **undoubtedly wrong**. Existing problem formulations in MARL (such as Dec-POMDPs or Stochastic Games) are agnostic towards the learning algorithm (i.e. including whether they are independent or centralized). As someone who has authored and reviewed papers in MARL, these models simply formalize the interaction between agents and the way their actions affect the state of the environment/other information the agents perceive (i.e. their rewards or observations included).
>
> Now, the use of MDPs to model interaction between agents in a multiagent system is simply incorrect (including the book referred to by the authors). This is because there is no "system" viewpoint in an MDP. MDP has always been designed for single-agent RL where each agent observes the entire state and gets one reward scalar (i.e. not multiple scalars like the authors' formulation) with the transition and reward function assumed to be Markovian. This Markovian and the single-agent assumption simply makes MDP unsuitable for modelling MARL.
>
> However, what the authors describe is more like a Stochastic Game [1] (importantly, not a Dec-POMDP since agents may have different reward functions). I highly urge the authors to reformulate their problem as this. I also encourage the authors to refer to the book from Shoham & Leyton-Brown (2008) to find different problem formulations that are more suitable for a multiagent setting.
>
> Citations:
>
> [1] Shapley, L.S. Stochastic Games. Proceedings of the National Academy Of Sciences, 39(10):1095–1100, 1953.
>
> [2] Shoham, Y. and Leyton-Brown, K. Multiagent Systems: Algorithmic, Game Theoretic, and Logical Foundations. Cambridge University Press, 2008.

---

> > ### Author Response · Authors · 2023-11-21
> > **Reply to Reviewer daQk's Response (1/4)**
> >
> > We really appreciate the reviewer's constructive and insightful feedback.
> >
> > **W1. Minor-Weakness: Citations**
> >
> > We thank the reviewer's advice and emphasized the theoretical aspect of our work as opposed to most ZSC/AHT works in our newly updated revision.
> >
> >
> > **W2. Minor-Weakness: Problem Formulation**
> >
> > - We fully agree with the reviewer that our problem formulation is  a stochastic game, particularly the **Stochastic game with independent learners**, i.e., $\langle \mathcal{X},\{\mathcal{U}\_A, \mathcal{U}\_H\}, P, \{r\_A, r\_H\} ,\gamma \rangle$ (as the same as in Definition 6.2.1 [R2]), where "each agent chooses best-response type actions to other agents' strategy without any coordination with her opponent" [R1], and in fact this is how we put together our formulation.  More specifically, as presented in Section 2.1 in page 4, AV chooses action by maximising the $L$-step lookahead planning objective
> > $$  \hat{\pi}(x(t) \vert {u\_H}) = \arg \max\_{u\_A(t)} \max\_{\{ u\_A(t+1),\cdots,u\_A(t+L-1) \}} \hat{Q}\_{t}(x(t),u\_A(t),\hat{u}\_H(t))$$
> >
> > And HV chooses actions by maximizing the $N$-step planning with bounded rationality, i.e.,
> >
> > $$ u\_H^{\*}(t):=\arg\max\_{u\_{H}(t)}\max\_{u\_{H}(t+1),\cdots, u\_{H}(t+N-1)} \Phi\_H(x(t),u\_A(t),u\_H(t))$$
> >
> > To improve the presentation, we have  revised the terminologies to use  **Stochastic Game** explicitly in **Section 2 Preliminary**  when introducing the problem setting in the new revision.  We also thank the reviewer for the reference book and have included it in our revision.
> >
> > - We apologize that in our previous response we had misunderstood your comments "go against MDP", so we had tried to
> >     clarify that the "our multi-agent formulation can be viewed as a generalization of *MDP frameworks*" (ref. Section 2.2.1[R3], Section 3 [R1]).  Specifically, in a Stochastic game with independent learners, each agent needs to solve a POMDP problem with its own reward function, and collectively  the agents constitute a multi-agent RL system with independent learners.
> >     We hope we are on the same page now.
> >
> >
> > - We fully agree with the reviewer that the MARL formulations per se "are agnostic towards the learning algorithms."  In our original response to "Minor-Weakness: Problem formulation", by saying "same algorithm", we meant that ``many MARL formulations  for  Dec-POMDP in theoretical studies, often assume all agents  use the same RL algorithms   for the sake of tractability,''  e.g.,
> >     -  All agents are assumed to use Q-learning while making decisions (ref. Eqn. (7)(8)) [R4].
> >     - Agents are assumed to choose the best response policy by maximizing the $Q$-function given history information (with the same form) (ref. Eqn. (4)) [R5].
> >     - The theoretical results are built upon the settings when all robots executes the same algorithm (ref. Algorithm 2 or Algorithm 3) [R6].
> >
> >     Our work is different from previous settings and consider the case where AV and HV use *different learning methods*, i.e., longer-term look-ahead planning and myopic decision making, to achieve their objectives.

---

> ### Comment · Reviewer_dAQk · 2023-11-21
> **Response to Author Rebuttal (2/4)**
>
> **W3. Major Weakness-AV and HV's policy is not adaptive?**
>
> I understand the authors' arguments. However, there seems to be a misunderstanding about what **adaptive** means. By adaptive, like Loftin and Oliehoek (2022), I am referring to agent policies that **change according to other agents' observed behaviour (i.e. based on observation-action history) from the past**. Formally, such adaptive policies can be formalized as $\pi: \mathcal{H} \rightarrow \Delta(A)$ (see background section from Loftin and Oliehoek (2022)).
>
> In comparison to the above concept **adaptive policies**, the policies used in this work's analysis are not adaptive since the decision-making of either human or AI agent is only based on the current state ($x_{t}$) and not the sequence of states and agent actions observed in the past (see that both agent and human objective when choosing actions are not conditioned on past information from timestep 0 to $t-1$). While there has been an effort to address this in the revision by conditioning on $u_{H}$, it is not clear how these previous actions translate into the objective for action selection (Expression 2) since they're not involved in this expression at all. Meanwhile, the human policy is not even assumed to depend on the previous actions of the agent.
>
> I believe it is important for the theoretical analysis to incorporate analysis of agents/humans that adapt their behaviour (i.e. having a policy that is dependent on the previous action of other agents). After all, humans change their driving policy according to the previous actions displayed by other drivers. For instance, if you see a driver who displays a tendency to be reckless, you may want to drive carefully around them. Meanwhile, you may want to overtake a driver that in the recent past tends to be slow and conservative. Without modelling the effects of different past actions to the agent's regret, it feels as if the applicability of the work will be very limited.
>
> (Adaptive Regret) Thanks for clarifying that the work is using adaptive regret. However, I think the writing has to be improved based on what I mentioned above. If $\pi_{A}^{t}$ is not defined as a policy conditioned on observation-action history, then its value on both terms in the expression will be the same. In that case, there will be no need to use adaptive regret at all (since the agent will not adapt to different policies when interacting with $\pi_{H}$ or $\pi_{H}^{t}$ ). In this sense, there has to be better clarity on how $u_{H}$ affects the policy of the agent in Expression 2. Again, currently, it does not appear as if $u_{H} $ affects anything in the RHS of Expression 2.

---

> ### Author Response · Authors · 2023-11-22
> **Reply to Reviewer dAQk's response (2/4)**
>
> **W3. AV and HV's policy is not adaptive?**
>
> **A3.** We thank the reviewer for clarifying your meaning of  "adaptive policy" in the recent comments. "Adaptive policy" can take many different forms, and our proposed adaptive policy'' is a widely used one in the literature (e.g., [R1, R2]).
>
> - ("Policies only depend on current state and not the sequence of states"?) First, the history state information has already embodied in the current state. Meanwhile, the polices depends **not only on** the current state $x\_t$ but also on the (predicted) other's actions (cf. Equation (2) and Equation (4)). Further, the policy would depend (indirectly) on the sequence of states and agent actions observed in the past through the current state. We model both AV and HV as an underactuated system such that the action of AV will have impact on HV, and vice versa. For instance, we have clarified in our original response that HV's policy depends on her prediction on AV's action $u\_A(t)$, $t=t+1,t+2,\cdots$.
>
> - ("Policies are not adaptive since it does not depend on the sequence of states and agent actions observed in the past"?) Our polices are indeed adaptive and follow the standard definition as in adaptive control and reinforcement learning in general.  We verbatim copy the definition from some of these papers below.
>
>     - (Adaptive Control) In the well-known textbook [R1] "adaptive control" by Karl J. Astrom et al., the adaptive controller is defined as "an adaptive controller is a controller with adjustable parameters and a mechanism to adjust parameters (in response to changes in dynamics)"
>     - (Reinforcement Learning) In the suvery paper [R1], "reinforcement learning refers to a class of methods that enable the design of adaptive controllers that learn online, in real time.". In particular, adaptation in reinforcement learning, is more about staying ahead of evolving challenges.
>
>      In our work, the adaptation of the agent's policy is achieved through the prediction on the future state (and actions of other agent). Clearly, the prediction of the future state can use end-to-end offline learning method  and/or previous collected history data. However, designing "an algorithm to predict the future state" and determine "what is the needed  information" are not the focus of our work.
>
>      In response to your suggestions on using history observations, we point out that it is well-known that solely using history information can be harmful in RL. For instance, the distribution shift issue in offline RL and the state-action visitation distribution shifts when using history polices in model-based RL.
>
> - ("Applicability of the work will be very limited"?) Our work is not restricted to any specific model on *how to use the history observation to predict the future*,  and in fact, it is a general setting that can serve as a basis for future studies. For instance, AV's prediction error of HV's future state by using historical data can be encapsulated in the term $\epsilon\_A$. In our work, we assume $\epsilon\_A$ to follow a Gaussian distribution. With explicit prediction model, it is feasible to derive the prediction error distribution conditioned on the history information.
> - ("agent will not adapt to different policies when interacting with $\pi\_H^t$"?) This is incorrect. For instance, in the defined reward function of AV, i.e., $  r_A(x,u\_A,u\_H)= f\_A(x,u\_H) + u\_A^{\top}S\_Au\_A$ (cf. Assumption 2), where the first term $f\_A$ is directly related to HV's policy. Based on this definition, AV's policy is derived by optimization the $L-$step lookahead planning, i.e., the summation of the future reward.
>
> [R1] "Adaptive Control" by Karl J. Astrom et al.
>
> [R2] Lewis, Frank L., Draguna Vrabie, and Kyriakos G. Vamvoudakis. "Reinforcement learning and feedback control: Using natural decision methods to design optimal adaptive controllers." IEEE Control Systems Magazine 32.6 (2012): 76-105.

---

### Official Review · Reviewer_BMkw · 2023-11-03

**Soundness:** 3 good
**Presentation:** 2 fair
**Contribution:** 3 good
**Rating:** 5
**Confidence:** 2

**Summary:**

This is a theory paper, analyzing a heterogeneous system of an automated vehicle (AV) and a human vehicle (HV) modeled as a joint MDP with continuous actions. The AV's model of the HV is N-step look ahead optimization plus Gaussian noise. Authors derived upper bounds for the following cases:
1. AV's performance Gap with linear dynamics
2. AV's performance Gap with non-linear dynamics
3. AV's Regret with non-linear dynamics
4. HV's Regret with non-linear dynamics
5. Joint HV-AV's Regret with non-linear dynamics

**Strengths:**

+ Great theoretical analysis that lays foundation for mixed human and autonomous systems
+ In addition to the upper bound derivation, authors plotted the upper bound for several conditions for better explanation of their impact

**Weaknesses:**

- Generalizability: Assumptions 1 and 2 can limit the applicability of the theory in practice. While I understand Assumption 2 has been widely used in the past, it would be great to discuss limitations these two assumptions would play in practice.
- Writing and clarity: there are writing issues with the paper (see minor comments for examples). Also I could not find the values for certain setups (e.g. settings 1-5).

Minor comments:
In definition of Q^π(x,u_A,u_H), the expectation needs \pi,
optimization on an proxy object => on a proxy
Let Let => Let
et al. (2022)Sadigh et al. (2016) => add ,

**Questions:**

Can elaborate on the limitations of your assumptions in practice?

---

> ### Author Response · Authors · 2023-11-15
> **Response to Reviewer BMkw**
>
> We thank reviewer BMkw for your thorough reading and positive feedback.
>
> **Q1. Limitations of Assumption 1 and Assumption 2**
>
> **A1.** We summarize the limitations of two assumptions as follows:
>
> - (**Assumption 1**: Function Approximation Error) In practice, since the underlying optimal value function is unknown, a commonly used approach to estimate the function approximation error $\epsilon\_{v,t}$ and $\mu\_{v,t}$ is to compare the difference between the rollout of the current policy (to estimate $\hat{V}\_t$) and Monte-Carlo Tree Search (MCTS) (to estimate $V^{\*}$) [R1]. However, in order to get an accurate estimation of the optimal value, MCTS need to be applied  to different policies and it can be time-consuming if the state space and action space are large. One of the promising approach is to leverage an offline dataset with the interaction history between the HVs and AVs [R2].
> -  (**Assumption 2:** Reward Structure) In order to obtain the reward parameters $S_H$ and $S_A$ for HVs and AVs, various factors may be taken into considerations, e.g., safety, speed, comfort. In practice, the reward function design is a long-standing open question and highly depends on the problem of interest. For instance, [R3] also considers reference path deviation to avoid the vehicles to drive out of the lane. Handcraft all the factors that matter to the questions can be challenging. A promising way to efficiently learn such a reward signal can be achieved by Inverse Reinforcement Learning (IRL) based on HVs and AVs driving data.
>
> [R1] Chen et al. "Randomized Ensembled Double Q-Learning: Learning Fast Without a Model." ICLR. 2020.
>
> [R2] Fu et al. "D4rl: Datasets for deep data-driven reinforcement learning." arXiv preprint arXiv:2004.07219 (2020).
>
> [R3] Ran et al. "Safety assurances for human-robot interaction via confidence-aware game-theoretic human models". ICRL. 2022.
>
> **Q2. Writing issues.**
>
>  **A2.** We thank the reviewer for pointing out the typos and the missing parameters setup and we have corrected them in our revision (all the revisions are marked in blue).

---

> ### Author Response · Authors · 2023-11-18
> **Thank Reviewer BMkw for your Review**
>
> Thank you Reviewer BMkw again for your detailed review.
>
> Since the final stage of the discussion between reviewers and authors will end soon, please let us know if you have any further comments on our response to your concerns, we will be more than happy to answer your questions.

---

> > ### Comment · Reviewer_BMkw · 2023-12-05
> >
> > Thank you for your response and updating the paper. I agree with other reviewers that presentation and clarity still remains a challenge for the paper. I believe a deep sweep in writing the paper and clarifying the raised concerns can boost the paper. I reduced my score by 1 point.

---

### Author Response · Authors · 2023-11-15
**Summary of Revisions**

We thank all the reviewers for their meticulous reading and constructive comments. We appreciate the positive comments from the reviewers, including "this work **provides a formalization of a problem that has not had extensive theoretical analysis**" (Reviewer Sgxf), "**offers a novel perspective on mixed autonomy**" (Reviewer DRuU), "**Novelty of regret analysis**" (Reviewer dAQk) and "**Great theoretical analysis that lays foundation for mixed human and autonomous systems**" (Reviewer BMkW).

We summarize a few major concerns raised by the reviewers and the corresponding revision we made below (all changes are marked in blue font in the revision).

- *Limitation of Assumption 1 and Assumption 2 (Reviewer **BMkw**).* We have added detailed discussion on the limitations in Section 3.1.

- *Clarification on the Adaptive Policy Assumption (Reviewer **dAQk**).* We clarify that the HV-AV interaction considered in this work indeed considers the adaptive nature in the real-world (ref. Section 2.1),  and elaborate further in Section 3.1, and Section 3.2 the connection between the regret definition in this work and the adaptive regret in the literature.

- *The significance (Reviewer **dAQk**)  and  robustness (Reviewer **DRuU**)*  of the problem formulation.   We provide a detailed comparison between our formulation and other  related multi-agent formulation in Section 2 and Appendix A. Meanwhile, we remark that our problem formulation in Equations (2) and (3) can be carried over to a variety of different learning methods.

- *Main take-away and detailed analysis from the theoretical results (Reviewer **Sgxf**).* We re-organize the analysis for the theoretical results and extract insight from the tightness of the bounds in Section 4. We also clarify further the main take-away, corroborated by  our empirical results in Figures 1-4.

In what follows, we provide point-to-point responses to each reviewer's comments.

---

> ### Author Response · Authors · 2023-11-22
> **Reply to All Reviewers: Summarize of Discussion**
>
> We thank all the reviewers for participating in the discussion and we appreciate your  constructive comments. We summarize the main concerns that proposed by reviewers at this stage and our response in the table below.  Please let us know if you have any other questions, we will be more than happy to clarify and share our thoughts.
>
> |  Main Concerns     | Our Response |
> | ----------- | ----------- |
> | **Reviewer dAQk**: he policies considered in this work "is not adaptive policies" in the sense that the policies should "change according to other agents' observed behaviour (i.e. **based on observation-action history**) from the past." as in [R0] ([R0] considers repeated matrix game)    | 1) We clarify Adaptive policy" can take many different forms, and our proposed "adaptive policy" is a widely used one in the literature (e.g., [R4, R5]). 2) We consider the **Stochastic game with independent learners** [R1], where "each agent chooses best-response type actions to other agents' strategy without any coordination with her opponent". We use the same setting as in previous work [R2, R3] to enforce the turn-taking, in which AV predicts the robot’s action prior to selecting her actions. It maintains the adaptive nature of two agent’s interaction since HV and AV can influence other agent’s policy, e.g., by selecting policies that "force" a particular response.       |
> |**Reviewer dAQk**: Assume HV and AV have fixed policy? Consider different autonomous vehicles learning algorithms?    | 1) We clarify that we do not have the assumption on fixed policies. 2)  We clarify that by saying "different learning strategies", we refer to the setting where **AV's learning strategy (or "learning algorithm") is different from HV's** during the interaction        |
> | **Reviewer Sgxf**:  The intuition of the bounds. | In the revision we have re-organized the interpretation based on our original manuscript for all the bounds provided  and highlighted the intuition. |
>
> ----
>
> We clarify the differences between our work and [R0] (as pointed out by Reviewer dAQk) in the table below.
>
> | Setting      | [R0] | Our Work |
> | ------ | ----------- |-------------- |
> |Problem Setting | Repeated Matrix Game | Stochastic (Markov) Game with Independent Learner|
> |Bounded Rationality |  &cross; | HV has bounded rationality|
> |Planning with prediction | &cross; | Both AV and HV make prediction on other agent's actions|
> |Heterogeneous Decision Making | &cross; | AV and HV are assumed to use different learning strategies |
> |Policy | Consider the past observations in the policy | The policy is conditioned on the  prediction of other agent's future actions. Both agents are modeled as an underactuated system. |
>
> ----
>
> In particular, our work conducts a comprehensive analysis on AV and HV's learning regret to answer the questions: 1) *How does the learning performance depend on HV's bounded rationality and AV's planning* We provide the the upper bound on the regret for both AV and HV; 2) *How do different decision making strategies impact the overall learning performance*? We examine the regret dynamics of the overall system theoretically and show how do different learning strategies between AV and HV affect the learning performance during each individual interaction through empirical study.
>
>
>
>
> [R0] Loftin \& Oliehoek. 2022. "On the Impossibility of Learning to Cooperate with Adaptive Partner Strategies in Repeated Games". ICML 2022
>
> [R1] Asuman et al., "Independent learning in stochastic games." arXiv preprint arXiv:2111.11743 (2021).
>
> [R2] Sadigh, Dorsa, et al. "Planning for cars that coordinate with people: leveraging effects on human actions for planning and active information gathering over human internal state." Autonomous Robots 42 (2018): 1405-1426.
>
> [R3] Sadigh, Dorsa, et al. "Planning for autonomous cars that leverage effects on human actions." Robotics: Science and systems. Vol. 2. 2016.
>
> [R4] "Adaptive Control" by Karl J. Astrom et al.
>
> [R5] Lewis, Frank L., Draguna Vrabie, and Kyriakos G. Vamvoudakis. "Reinforcement learning and feedback control: Using natural decision methods to design optimal adaptive controllers." IEEE Control Systems Magazine 32.6 (2012): 76-105.

---

### Meta-Review · Area_Chair_yCyK · 2023-12-06

**Metareview:**

Summary: This paper studies mixed autonomy settings—with autonomous vehicles and human vehicles. It specifically focuses on models where the human is boundedly-rational and the autonomous vehicle makes decisions in a receding-horizon fashion by relying on predictions of the human. Under this model, the paper proves regret bounds from each agent’s point of view and also shows empirical results on how the AV’s planning horizon affects regret.

Strengths: The paper presents a novel regret analysis for human-AV interaction where humans have bounded rationality and where the AV predicts the humans, and the paper presents some numerical results in addition to the primary theoretical results.

Weaknesses: Several reviewers raise concerns about the clarity of the manuscript and the stated results. I reviewed the paper myself and while I believe that overall the paper is reasonably well written, I do agree with the reviewers that the positioning, assumptions, and results could be more clearly stated. Reviewers also raised questions about if the paper's assumptions were realistic. For example, the paper only focuses on two-agent interactions while autonomous driving interactions require N>2 agents--what are the implications of this approach for such scenarios? Another assumption is that the AV's predictions of the HV are limited to unimodal predictions centered at the ground-truth action with Gaussian uncertainty. In reality, 1) multimodal predictions are key for capturing real-world interactions and are SOTA in autonomous driving, and 2) getting noisy access to the true future actions over the L-step horizon is a very strong assumption. For this reason, the manuscript would be strengthened by making this assumption clear and discussing how it helps gain insight into the setups one may see in reality. Since the paper mentions a finite horizon planning for the human and AV planning, it would also be nice to clarify how frequently the human and the AV are expected to re-plan or re-predict within this receding-horizon model --- this re-planning or re-prediction frequency is key for real-world performance and if the theory could shed light on how the regret is affected as a function of this frequency this would be very interesting to practitioners and theorists alike.

**Justification For Why Not Higher Score:**

I recommended rejection at this time since the manuscript could be presented more clearly and the assumptions under which the results hold warrant further discussion and investigation to make a bigger impact (e.g., unimodal predictions centered at true future human behavior, how frequently do the agents re-plan or re-predict).

**Justification For Why Not Lower Score:**

N/A

---

### Decision · Program_Chairs · 2024-01-16

Reject